# Disentangling Sources of Future Uncertainties for Water Management in Sub-Saharan River Basins

Alessandro Amaranto [1], Dinis Juizo [2], and Andrea Castelletti [3,1]

[1]Department of Electronics, Informations and Bioengineering, Politecnico di Milano, Piazza Leonardi da Vinci, Milano, Italy.
[2]Department of Civil Engineering, Eduardo Mondlane University, Maputo, Mozambique
[3]Institute of Environmental Engineering, ETH Zurich, Zurich, Switzerland

**Correspondence:** Andrea Castelletti (casandre@ethz.ch)

**Abstract.** Water management in sub-Saharan African river basins is challenged by uncertain future climatic, social and economical patterns, potentially causing diverging water demands and availability, as well as by multi-stakeholder dynamics, resulting in evolving conflicts and tradeoffs. In such contexts, a better understanding of the sensitivity of water management to the different sources of uncertainty can support policy makers in identifying robust water supply policies balancing optimality and low vulnerability against likely adverse future conditions. This paper contributes an integrated decision-analytic framework combining optimization, robustness, sensitivity and uncertainty analysis to retrieve the main sources of vulnerability to optimal and robust reservoir operating policies across multi-dimensional objective spaces. We demonstrate our approach onto the lower Umbeluzi river basin, Mozambique, an archetypal example of sub-Saharan river basin, where surface water scarcity compounded by substantial climatic variability, uncontrolled urbanization rate, and agricultural expansion are hampering the Pequenos Lipompos dam ability of supplying the agricultural, energy and urban sectors. We adopt an Evolutionary Multi-Objective Direct Policy Search optimization approach for designing optimal operating policies, whose robustness against social, agricultural, infrastructural and climatic uncertainties is assessed via robustness analysis. We then implement the GLUE and PAWN uncertainty and sensitivity analysis methods for disentangling the main challenges to the sustainability of the operating policies and quantifying their impacts on the urban, agricultural and energy sectors. Numerical results highlight the importance of robustness analysis when dealing with uncertain scenarios, with optimal-non robust reservoir operating policies largely dominated by robust control strategies across all stakeholders. Furthermore, while robust policies are usually vulnerable only to hydrological perturbations and are able to sustain the majority of population growth and agricultural expansion scenarios, non-robust policies are sensitive also to social and agricultural changes, and require structural interventions to ensure stable supply.

## 1 Introduction

The availability of freshwater is a limiting factor to food production, energy generation, and industrial consumption around the globe (Hermoso, 2017; Zampieri et al., 2018). Investing in new infrastructure is still the predominant option to expand storing and conveying capacity, particularly in sub-Saharan Africa and South-east Asia that have the larger untapped potential (Fields et al., 2009). Yet, this hard path water solution is evoking contentious debates for the considerable environmental and

social costs of damming rivers (Moran et al., 2018), to a point that efficient operation of existing infrastructure, rather than planning new ones, is becoming critical to balance tradeoffs between supply and demand (Gleick and Palaniappan, 2010). In sub-Saharan Africa, more than 2000 dams have been built and over 200 are currently under construction to enhance food security and increase hydropower production (Kibret et al., 2016). However, projected change in climate, population growth and agricultural patterns will likely challenge the ability of existing and planned dams to produce the level of benefits that

triggered the investment for their construction (Giuliani et al., 2016b). Understanding the impact of those uncertainty sources on reservoir operation is therefore key for developing robust operating policies that support policy makers towards sustainable river basin management.

An archetypal example of a highly regulated, fast evolving South-saharan hydrosystem is the Lower Umbeluzi river basin, Mozambique. About 45 km upstream of its delta in Maputo bay, the river flows in the Barragem de Pequenos Lipompos

reservoir, which is operated to balance hydropower production, urban supply to the two million inhabitants of Maputo province, and irrigation supply of the 3600 ha of agricultural districts, mostly growing tropical fruits and sugarcane. A still ongoing five-year long drought has boosted crop prices by about 50%, hindering food access to a population currently growing at rate of 0.6% per year and exacerbating conflicts among the urban, agricultural and energy sectors. Droogers et al. (2014) estimated a further annual increase of 2% in the urban population and of 2% in irrigated area for the coming decades, while a projected

10% climate-change induced reduction in precipitations seriously endangers the existing fragile equilibrium among sectors. To cope with the expected growth in water scarcity, the World Bank is funding the Greater Maputo Water Supply Expansion Project: a sequence of infrastructural interventions aimed at supplying the city of Maputo with an additional inflow from the Sabie river basin. Started in 2013, the project is expected to be completed in the following years. The uncertain evolution of climatic, agricultural, infrastructural and social patterns in the area calls for assisting policy makers in: (1) the development of

robust reservoir operating policies; (2) a deep understanding of the main sources of vulnerability challenging the sustainability of water supply strategies; and (3) a quantitative assessment of the impact of such uncertainty sources across the agricultural, the energy and the urban sectors.

To do so, we implement an integrated decision-analytic framework combining optimization, robustness, sensitivity and uncertainty analysis with the threefold objectives of (1) designing operating policies that integrate optimality with low vulnera-

bility against likely adverse future conditions; (2) retrieving the main sources of vulnerability to water infrastructure operation across a multidimensional objective space; and (3) identifying the evolution in the system drivers which causes a policy to become unsustainable.

The framework is composed of an optimization routine based on Evolutionary Multi-Objective Direct Policy Search (EMODPS) to design optimal control policies, whose robustness against deeply uncertain (i.e. with unknown probability of occurrence)

climate, socio-economic and infrastructural scenarios is then tested by means of robustness analysis. To discover the main uncertainty sources and apportion their specific impact on the hydrological system, we employ the Generalized Likelihood Uncertainty Estimation (GLUE, Beven and Binley 1992) and the PAWN density-based sensitivity analysis (SA) method (Pianosi and Wagener, 2015), respectively.

The main sources of uncertainty we consider are: the projected increase in water demand following urbanization (population uncertainty) and irrigation development (agricultural uncertainty) in the area; the magnitude of streamflow depletion due to climate change (climatic uncertainty); and the completion date of the greater Maputo water supply expansion project (infrastructural uncertainty). The proposed methodology builds upon recent studies in the field of many-objective reservoir operation (Giuliani et al., 2016b; Denaro et al., 2017; Giuliani et al., 2019; Zaniolo et al., 2018, 2019) and of multi-objective robust decision making (Giudici et al., 2020; Herman et al., 2015), by employing SA to investigate the role of uncertain exogenous drivers in shaping the effectiveness of optimal control policies across multiple, competing sectors. So far. even though it has been recognized that optimal planning and control methods should employ SA to identify water resources system vulnerabilities to both structural and parametric uncertainties (Herman et al., 2019), only few studies developed quantitative analyses to support water resource planning (e.g. Herman et al., 2015; Trindade et al., 2017, 2019; Groves et al., 2019). To the best of the authors knowledge, the only application coupling SA with reservoir operation and control problems is represented by Quinn et al. (2019), where SA was employed with the aim of understanding how optimized nonlinear control policies use endogenous information (reservoir state) to prescribe releases in a multi-reservoir multi-objective context. Our framework complements the findings by Quinn et al. (2019) by investigating the role of uncertain exogenous drivers in shaping the effectiveness of optimal operating policies to sustain the agricultural, urban and energy sectors.

## 2  Study Area Description

### 2.1  The Umbeluzi river basin

The Umbeluzi river flows across three countries (South Africa, Swaziland, and Mozambique), draining an area of about 5400 $km^2$ (Fig 1a) before discharging in the Indian Ocean through the Espirito Santo Estuary south of Maputo. (Juizo and Hjorth, 2009). The hydroclimatic regime is subtropical, with hot and wet summers from November to May, followed by dry, warm winters from June to October (Figures 2a and 2b). The average annual streamflow in the lower Umbeluzi (regulated through the Mnjoli Dam in Eswatini, in the upstream part of the river basin) is of about 220 $Mm^3$ (corresponding to 7 $m^3 s^{-1}$) and, following the hydroclimatic pattern, is unevenly partitioned through seasons, with 78% of the total discharge occurring in summer (171 $Mm^3$, corresponding to 9.3 $m^3 s^{-1}$) and the remaining 22% in winter (49 $Mm^3$, 3.75 $m^3 s^{-1}$). As far as the inter-annual variability is concerned, the Umbeluzi river basin is characterized by frequent prolonged droughts, with the average inflow reduced to only 119 and 100 $Mm^3$ in 2007 and 2015, respectively. Excluding the year 2000, where a terrible flood hit Mozambique causing about 800 fatalities, the wettest year among those on record (2006-2016) was 2010, when the total discharge volume reached about 360 $Mm^3$.

About 40 km upstream of the Estuary, the river flows into the Pequenos Libombos Dam (BPL). The dam has a storage capacity (including 10.2 $Mm^3$ of inactive storage) of 382 $Mm^3$. BPL was constructed in 1987 with the goal of supplying water to the metropolitan area of Greater Maputo (including the municipalities of Maputo, Matola and Boane), especially during winter, when the system is often exposed to droughts. Other operation targets include hydropower generation and irrigation supply, both upstream and downstream the reservoir. Upstream irrigation districts extend for about 2500 $ha$, and

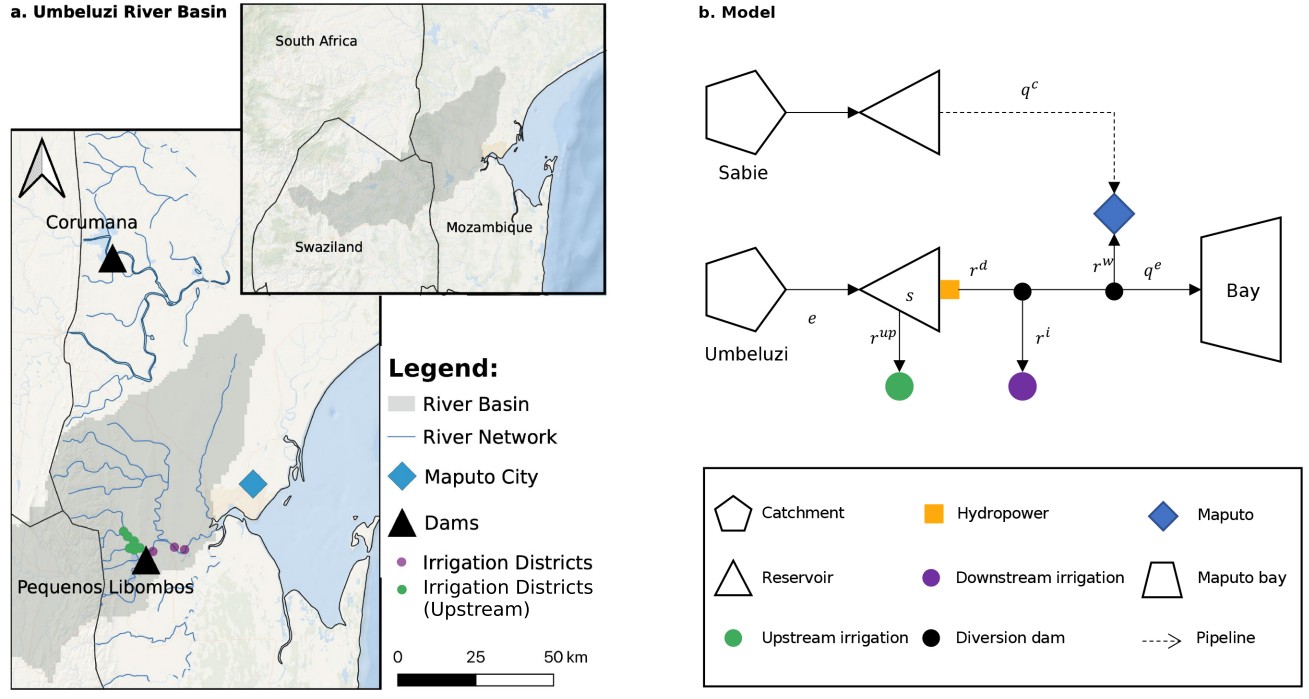

**Figure 1.** (a) Study Area. (b) Topological map of the system.

have an yearly water demand of 22.8 $Mm^3$ (Figure 2c). They abstract water directly from the reservoir to grow mainly tropical fruits (mango and bananas). Water is discharged from the dam into a power plant with a capacity of 1.8 MW. After flowing through the turbine, reservoir releases serve both urban and irrigation (bananas and sugarcane) demands, which are estimated around 80 and 11.5 $Mm^3 y^{-1}$ (Figure 2c), respectively. To preserve ecosystem sustainability, a minimum flow constraint corresponding to 15% of the cyclostationary monthly inflow is imposed in the estuary. A summary of the main hydroclimatic patterns across seasons, including aggregated values of water demands by sectors, is provided in Table 1.

**Table 1.** Summary of the main hydroclimatic variables and water demand sources aggregated by sector

|  | Total $[Mm^3]$ | Summer $[Mm^3]$ | Winter $[Mm^3]$ |
|---|---|---|---|
| Inflow | 220 | 171 | 49 |
| Upstream Irrigation | 22.8 | 12.1 | 10.7 |
| Downstream Irrigation | 11.5 | 6.1 | 5.4 |
| Environment | 33 | 25.65 | 7.35 |
| Urban | 80 | 46.6 | 33.4 |

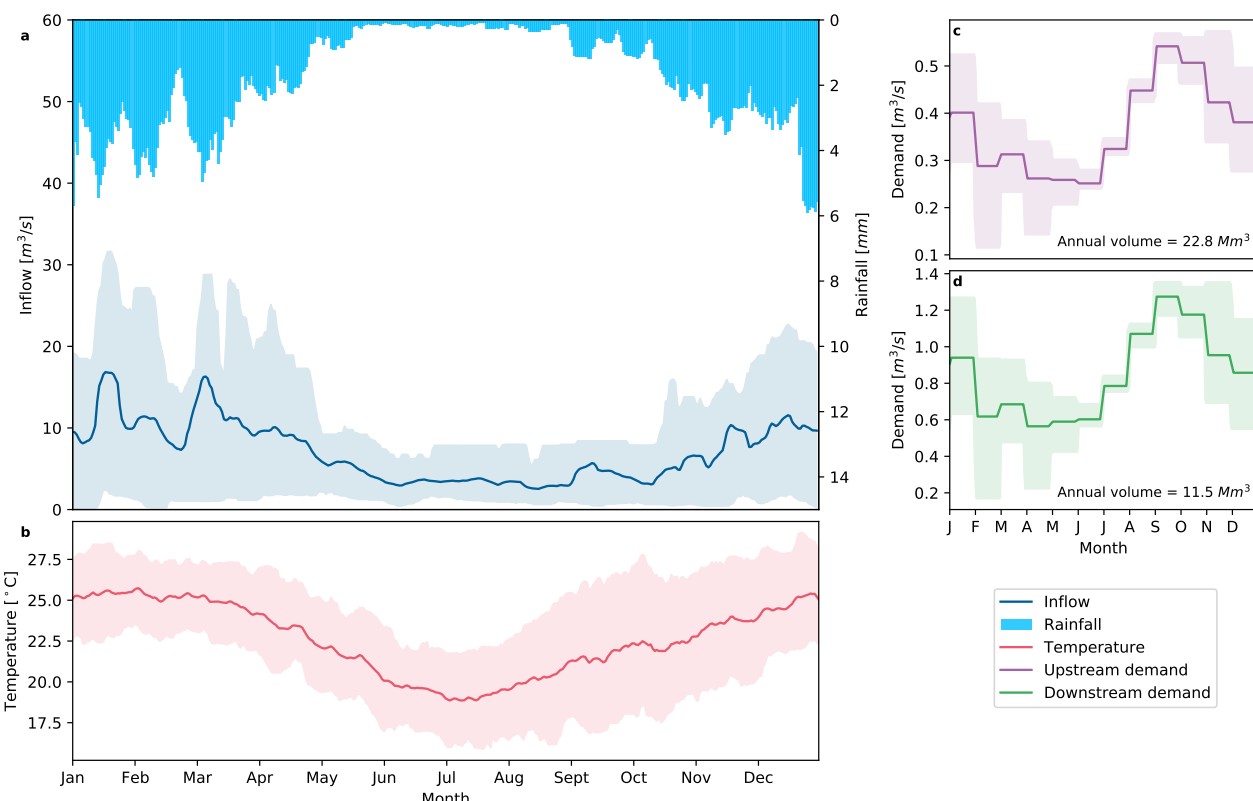

**Figure 2.** Ciclostationary average of the main hydroclimatic variables computed over a 10-days moving window. (a) Precipitation and streamflow; (b) temperature; (c) downstream irrigation demand; and (d) upstream irrigation demand. The shaded areas represent the 10-90th interquantile range for each variable.

Recently, a remarkable decrease in rainfall frequency and intensity has caused the reservoir storage to drop up to less than one fifth of its maximum capacity, forcing local authorities to suspend distribution for irrigation in 2016 (Macauhub, 2016) in order to ensure continuity in urban supply. This dry pattern is expected to be further exacerbated by climate change in the coming years, with an estimated precipitation decrease of about 10% and a temperature increase of $3°C$ (Droogers et al., 2014).

To mitigate the effect of frequent and prolonged drought episodes hitting southern Mozambique, the World Bank have recently financed the Greater Maputo Water Supply Expansion project (GMWSEP, Miguel (2019)). The project consists in a set of infrastructural interventions, mainly constituted by a water treatment plant downstream the Corumana dam, in the adjacent Sabie river basin, and by a 95 km pipeline connecting Corumana with the city of Maputo, ensuring an additional water supply capacity of $q^C$ of $1.8 \ m^3 s^{-1}$ (about 70% of the current urban water demand). The GMWSEP is expected to play a key role not only in mitigating drought effects but also in sustaining the rapidly increasing population of Maputo.

## 2.2 Umbeluzi model

We model the Umbeluzi river system (Fig. 1b) using a combination of conceptual and empirical models assuming a daily time
step for both physical processes and decisions. The BPL dynamics are described by the mass balance equation of the water
volume $s_t$ stored in the reservoir, i.e.

$$s_{t+1} = s_t + e_{t+1} - r_{t+1} \tag{1}$$

where $e_{t+1}$ is net inflow to the reservoir in the interval [t; t + 1] (i.e., inflow minus evaporation and seepage losses) and
$r_{t+1}$ is the volume released in the same interval. This is further decomposed into $r_{t+1}^d$ and $r_{t+1}^{up}$, representing the downstream
releases for hydropower production, urban and irrigation supply, and the upstream pumping to the upstream irrigation districts,
respectively. The actual releases $r_{t+1}^d = f_1(s_t, u_t^d, u_t^{up}, e_{t+1}, t)$ and $r_{t+1}^{up} = f_2(s_t, u_t^{up}, u_t^d, e_{t+1}, t)$ are formulated according to
the nonlinear, stochastic relations $f_1(\cdot)$ and $f_2(\cdot)$ between $r_{t+1}^{up}$, $r_{t+1}^d$ and the release decisions $u_t^{up}$ and $u_t^d$ (Piccardi and
Soncini-Sessa, 1991). The latter are in fact constrained by physical constraints (i.e. spillway activation and inactive storage
threshold) within a discretionary operating space by the maximum and minimum feasible release function (see Soncini-Sessa
et al. 2007 for more details). In particular, the minimum release function constrains the release to zero in case the available
volume in the reservoir $s_t$ equals the inactive storage $(10.2 Mm^3 s^{-1})$, while the maximum hydraulic outflow $r_{t+1}^{MAX}$ (including
both upstream and downstream releases) can be formulated as follows:

$$r_{t+1}^{MAX} = 4.6(h_t - 15.25)^{0.5} \tag{2}$$

Such constraints, in turn, implies interdependence among the release decisions, so that the total release never exceeds the
feasible release. We assume here feedback operating rules, parametric in $\zeta$, where decisions are conditioned upon the current
system conditions i.e.

$$u_t^{up} = f(t, I_t, \zeta) \tag{3a}$$

$$u_t^d = g(t, I_t, \zeta) \tag{3b}$$

where $I_t$ represents the information upon which the policy is based. The analytical expression for the functions $f$ and $g$ depends
upon the optimization problem formulation, and it is provided in section 3.2.2.

The aggregated downstream irrigation supply is modelled by means of a diversion dam, represented mathematically with an
empirical exponential function (Celeste and Billib, 2009) parametric in $\alpha$ and $\beta$ of the form:

$$u_{t+1}^i = \begin{cases} \min\left(W_t^d \cdot \left(\frac{r_{t+1}^d}{\alpha}\right)^\beta, r_{t+1}^d\right) & \text{if } r_{t+1}^d \leq \alpha \\ \min\left(r_{t+1}^d, W_t^d\right) & \text{else} \end{cases} \tag{4}$$

where $W_t^d$ is the aggregated downstream irrigation demand (Figure 2d). Equation 4 defines the fraction of releases to be
diverted for irrigation purposes as inversely proportional to $\alpha$, and exponentially growing with respect to $\beta$. It follows that,

according to the values assumed by such parameters, urban supply could (as it occurs, for example in case of $\alpha > r_{t+1}^d$ or for $\beta > 1$ when $\frac{\alpha}{r_{t+1}^d} < 1$) or could not ($\alpha < r_{t+1}^d$) be placed in the foreground with respect to irrigation. Even though this does not correspond to the actual operating rule of the diversion dam (which systematically prioritizes urban supply in case of water scarcity conditions), this study aims at exploring the whole irrigation-urban supply tradeoff. Therefore, equation 4 is set such that also solutions that favour irrigation could be discovered.

The periodic sequence of the control laws (over a period of 365 days) described in equations 3 and 4 constitutes the control policy $\pi_\theta$, where $\theta = |\zeta, \alpha, \beta|$ represents the vector of the control policy parameters.

The diversion rules allow hedging the water abstractions to account for downstream users (i.e. the city of Maputo), and the actual diverted flows $r_{t+1}^i$ are constrained by the the environmental flow requirement in the Espirito Santo Bay $q_t^e$ as follows.

$$r_{t+1}^i = \min\left(u_{t+1}^i, \left(r_{t+1}^d - q_t^e\right)^+, q^{max}\right) \tag{5}$$

where $q^{max}$ is the maximum diversion channel capacity.

Finally, the amount of water to be diverted for urban supply to the city of Maputo $r_{t+1}^w$ is computed as:

$$r_{t+1}^w = \min\left(\left(r_{t+1}^d - r_{t+1}^i - q_t^e\right)^+, W_t^w\right) \tag{6}$$

where $W_t^w$ is the urban demand.

The data necessary for the implementation of the analysis for the time-period December 2016-January 2006 were provided by The Administracão Regional de Ãgua (ARA) Sul, which is the water agency responsible for river basin management in southern Mozambique (including the Umbeluzi).

## 3  Methods and Tools

### 3.1  Integrated Decision-Analytic Framework

The integrated decision-analytic framework we adopted is represented in Figure 3. The approach is composed by three main blocks: optimization (O), robustness (R), and sensitivity-uncertainty (SU).

Block O is responsible for generating operating policies which are optimal under historical conditions. Block R extracts the optimal policies that are also robust against future changes in climate (climatic uncertainty), irrigation demand (irrigation demand uncertainty), infrastructures (infrastructural uncertainty), and urban demand (population uncertainty); block SU bridges robust policies (RP) with the operating objectives variability in response to uncertain input realizations, and allows the identification of the main sources of vulnerability for the hydrosystem.

The three blocks are interconnected as follows: first, to explore tradeoffs among stakeholders under the historical climatic, agricultural, infrastructural and urban demand drivers value (i.e., the baseline), we define the operating objectives, and run a multi-objective evolutionary algorithm to identify the optimal operating policies via optimization-based simulation. Then, we assess their robustness with respect to the future evolution of the drivers by perturbing the baseline across all the four (climatic,

agricultural, infrastructural and population) uncertainty dimensions to generate the states of the world (SOWs). We then re-simulate the system iteratively perturbed by the SOWs for each of the optimal operating policies identified via optimization and compute the worst objective function values. The robust policy for each stakeholder is subsequently identified as that yielding

the best performances in the worst condition (*minimax* robustness metric). To disentangle the sources of vulnerability for RP, we implemented an uncertainty and sensitivity analysis framework. In particular, uncertainty analysis is employed to quantify the objective function variability in response to the uncertain future evolution of the system's drivers. Here, following the well known definition of behavioral parameters (Beven and Binley, 1992; Montanari, 2005), we identify the behavioral system perturbations as those SOWs satisfying predefined performance requirements, i.e. those yielding acceptable objective function

values. Sensitivity analysis is responsible for determining the relative contribution of each individual uncertainty source in shaping the objective space and for ranking them across policies and objectives by means of a sensitivity index.

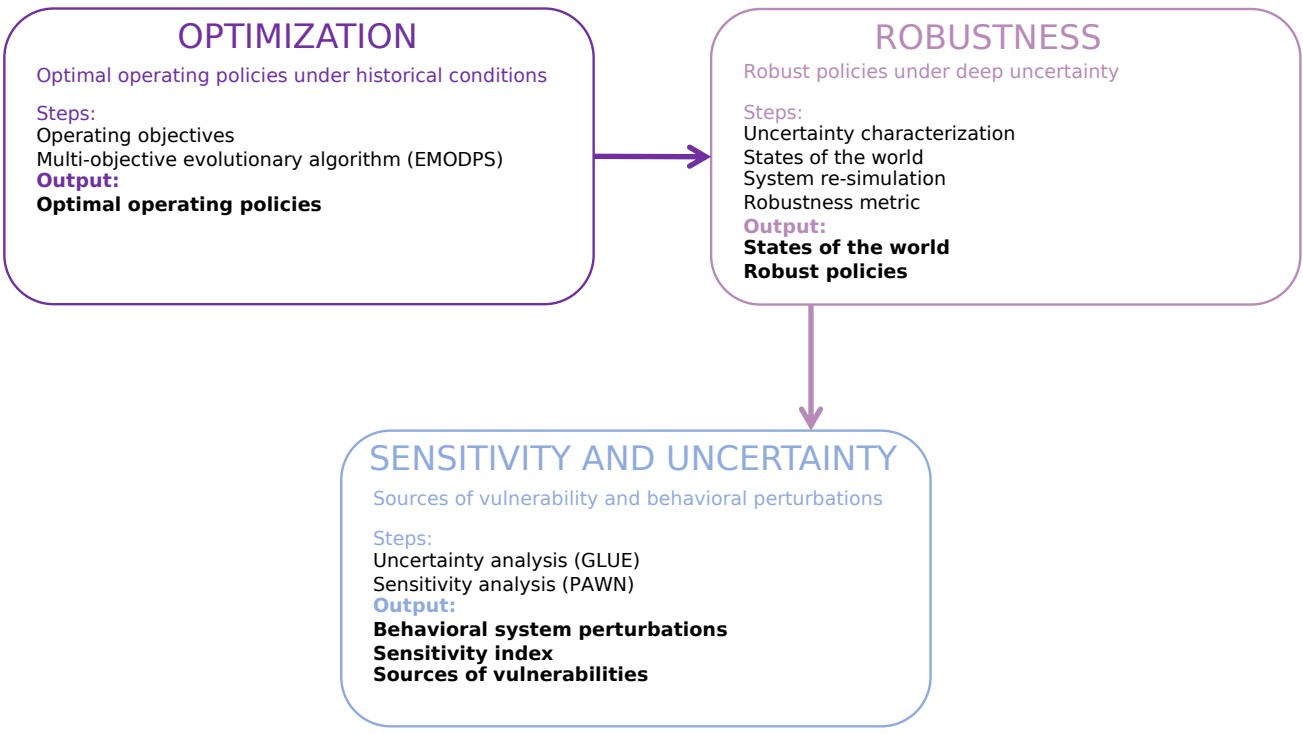

**Figure 3.** The integrated decision-analytic framework adopted in this study. Bold characters identify the output of a certain block, entering in the subsequent.

Further details about each block in Figure 3 are provided in the following sections.

### 3.2 Optimization

#### 3.2.1 Operating Objectives

We model the stakeholdes affected by the operation of the BPL dam through the following four utility functions:

1. Upstream irrigation deficit $J^{IU}$, expressed as the square difference between water supply and demand (to be minimized):

$$J^{IU} = \frac{1}{N_y} \sum_{t=1}^{N} \left( b_t \left( W_t^{up} - r_{t+1}^{up} \right)^+ \right)^2 \tag{7}$$

Where $N$ (days) is the simulation horizon, $N_y$ are the number of years in the simulation horizon, and $b_t$ is a weight
representing higher losses when the deficit occurs during the growing season. $W_t^{up}$ and $r_{t+1}^{up}$ are the irrigation demand and the amount of water pumped from the reservoir to upstream irrigation, respectively. The quadratic water supply deficit has been a traditional formulation in reservoir operations since the work by Hashimoto et al. (1982). The square of the irrigation deficit accounts in fact for crop vulnerability by penalizing higher shortages, which are more likely to compromise the crop growth, with respect to more frequent but smaller shortages, which are less dangerous to the crops.

2. Downstream irrigation deficit $J^{ID}$ defined similarly as in equation 7 (to be minimized):

$$J^{ID} = \frac{1}{N_y} \sum_{t=1}^{N} \left( b_t \left( W_t^{d} - r_{t+1}^{i} \right)^+ \right)^2 \tag{8}$$

where $W_t^{d}$ and $r_{t+1}^{i}$ are the irrigation demand and the reservoir releases diverted for downstream irrigation, respectively.

3. Urban deficit for the city of Maputo $J^{UD}$, computed as the difference between urban supply and demand (to be minimized):

$$J^{UD} = \frac{1}{N_y} \sum_{t=1}^{N} \max \left( \left( W_t^{w} - r_{t+1}^{w} - q_t^{C} \right), 0 \right) \tag{9}$$

where $W_t^{w}$, $r_{t+1}^{w}$ and $q_t^{C}$ are the urban demand, the reservoir release diverted for urban supply and the additional inflow from Corumana, respectively.

4. Hydropower production in the BPL power plant $J^{HP}$ (to be maximized):

$$J^{HP} = \frac{1}{N_y} \sum_{t=1}^{N} HP_t \tag{10a}$$

where $HP_t$ is the hydropower production [MWh] on day $t$, defined as:

$$HP_t = \eta g \gamma_w h_t r_{t+1}^{d} \cdot 10^{-6} \tag{10b}$$

where $\eta$ is the turbine efficiency (70%), $g = 9.81 \ m/s^2$ is the gravitational acceleration, $\gamma_w = 1000 \ kg/m^3$ is the water density, $h_t$ is the net hydraulic head and $r_t^{d}$ is the turbined flow.

### 3.2.2 EMODPS

The optimal operating policies under the baseline are designed using Evolutionary Multi-Objective Direct Policy Search (EMODPS) (Giuliani et al., 2016b). EMODPS is a simulation-based optimization approach, which has been recently demonstrated (Giuliani et al., 2016b) to successfully overcome the major limitations associated with traditional Stochastic Dynamic Programming (i.e. curses of dimensionality, modelling and multiple objectives) and derivatives. It is constituted by three main modules: (1) direct policy search (DPS); (2) nonlinear approximating networks; and (3) multiobjective evolutionary algorithms
(MOEA).

DPS is employed to explore the parameter space $\theta = |\zeta, \alpha, \beta|$ of the system operating policy $\pi_\theta$ that optimizes the expected long term cost, i.e.:

$$\pi_\theta^* = \underset{\theta}{arg\min}\,(\mathbf{J}_\theta) \tag{11a}$$

*where:*

$$\mathbf{J}_\theta = |J^{IU}, J^{ID}, J^{UD}, -J^{HP}| \tag{11b}$$

*subject to equations 1 to 6*

where finding $\pi_\theta^*$ means finding: (1) the optimal parameters $\zeta^* \in Z$ of the BPL reservoir operating policy; and (2) the optimal parameters $[\alpha^*, \beta^*] \in \Theta_{irr}$ for the regulation of the irrigation diversion canal. The parameters are intended optimal with respect to the objectives $\mathbf{J}_\theta$. The reservoir operating policy is selected such that policy inputs $I_t$ can provide information
feedback for the upstream and downstream release decisions $u_t^{up}(I_t), u_t^d(I_t)$. In this study, it is represented with a nonlinear approximating network of the Gaussian radial basis function family (RBFs), which are known for their generalization ability and robust performances in validation (Giuliani et al., 2016b; Quinn et al., 2019).

Mathematically, the operating policy can be expressed as:

$$u_t^{k=|up,d|} = \sum_{i=1}^{n} w_i^k \exp\left[-\sum_{j=1}^{m}\left(\frac{I_{(t),j} - c_{i,j}}{b_{i,j}}\right)^2\right]. \tag{12}$$

Where $n$ is the number of RBFs, $w_i^k$ is the weight of each RBF, $m$ is the number of inputs, and $c$ and $b$ are the center and radii of the RBF. The reservoir operating policy parameter vector is therefore constituted as: $\zeta = [w_i^k, c_{i,j}, b_{i,j}]$, and the number of parameters $n_\zeta$ to be found is $n(2m + k)$.

The input vector $I_t = |e_t, s_t, sin(2\pi t/365), cos(2\pi t/365)|$ includes the previous day inflow to the reservoir $e_t$, the storage in the reservoir $s_t$ and time $t$, which is here represented by a combination of sine and cosine functions: $sin(2\pi t/365)$ and
$cos(2\pi t/365)$.

Following Giuliani et al. (2016a), the number $n$ of RBFs is set to $m+k+1 = 7$. In addition to those of the reservoir operating policy, also the two parameters $\alpha$ and $\beta$ of the power law employed to approximate the diversion dam (as in equation 4) needs to be optimized. Following the above, the dimension of the parameter space equals $n(2m + k) + 2 = 72$.

In order to explore the parameter space and discover optimal values, we employ Multi-Objective Evolutionary Algorithms. The term evolutionary refers to the natural randomized mating, selection, and mutation processes that are mimed by the algorithms to evolve a Pareto-approximate set of solutions (Deb, 2001; Coello et al., 2007). MOEAs have proved to successfully deal with complex multiobjective optimization problems, including water reservoir operations (Maier et al., 2014). In this study, we employ the self-adaptive Borg MOEA (Hadka and Reed, 2013), which has been shown to guarantee high robustness in solving a variety of multiobjective problems when compared to other MOEAs (Salazar et al., 2016).

## 3.3  Robustness

We perform a robustness analysis with the aim of evaluating the robustness of the various operating policies identified via EMODPS over an ensemble of future realizations of the climatic, agricultural, infrastructural and urban demand drivers. According to Herman et al. (2015), a robustness analysis is usually carried out by performing the following sequential steps: (1) generation of alternative policies; (2) sampling of possible future scenarios and; (3) computation of robustness metric via system re-simulation. Table 3 provides a conceptualization of the three aforementioned steps tailored upon this study, following the well known XLRM framework (Lempert, 2003). In the framework, X are the exogenous uncertainty sources; L (of lever) are the different alternative water management strategies (i.e., the policies identified via optimization) to be explored; M (of measure) refers to the performance metrics used to rank the desirability of the different policies (L) in the face of the exogenous uncertainties (X); and, finally, R refer to 'relationships in the system' (i.e., the model), which define how the exogenous uncertainties (X), policies (L) as well as outcomes (M) are tied together and relate to each other (Ciullo et al., 2019).

**Table 2.** Conceptualization of the robustness analysis implemented in this study.

| Uncertain Factors [X] | Policies [L] |
|---|---|
| Climate | Optimal operating policies found via optimization |
| Irrigation demand | |
| Additional inflow from Corumana dam | |
| Urban water demand | |
| **Relationship [R]** | **Performance Metrics [M]** |
| Umbeluzi model | *minimax* robustness metric |

In other words, the alternative operating policies are those identified via optimization, while the generation of future scenarios is performed by perturbing historical trajectories assuming independent uniform distribution for the perturbation multipliers (Pianosi and Wagener, 2016). The multiplier range is either defined by a-priory expert knowledge of the system or based on experimental results. Climatic, agricultural, infrastructural and population scenarios are then combined to generate a set of uncertain states of the world. Then, the Umbeluzi model is used to re-simulate the system for each policy across all the SOWs. Each simulation produces a performance metric, used to assess the robustness of each policy. Among the robustness metrics

available in the literature (for a review see McPhail et al., 2018), we select *minimax*, a metrics that identifies the solution providing the best performance assuming the realization of the worst conditions.

More details on the formulation of states of the world and on the robustness metrics are provided below.

### 3.3.1 Formulation of the States of the World

We divide the future sources of uncertainty into four categories: (1) climatic, (2) agricultural, (3) infrastructural and (4) social. They are driven by uncertainty in (1) streamflow, (2) irrigation demand, (3) additional inflow from Corumana dam, and (4) population growth rate, and are here characterized as follows:

1. **Climatic uncertainty**: We generate high resolution scenarios of rainfall and temperature for the Umbeluzi river basin to the year 2100 by the quantile-quantile mapping (QQ-Mapping) downscaling procedure. We apply QQ-Mapping to coarse resolution data from three different Regional Circulation Models (ICHEC RCA4, ICHEC RACMO and ICHEC HIRHAM5, developed by the Swedish Meteorological and Hydrological Institute, the Royal Netherlands Meteorological Institute, and the Danish Meteorological institute), simulated over three distinct representative concentration pathways: the RCP 2.6, RCP 4.5 and RCP 8.5 (Field, 2014). We use the resulting nine precipitation and temperature trajectories to force an HBV model (Lindström et al., 1997) validated over the control period, generating nine inflow trajectories. We then use the minimum (0.05) and the maximum (0.4) of the projected percentage inflow decrease as the feasibility set boundary of an uniform distribution, from which we extracted $K = 10000$ inflow multipliers in the interval [0.6, 0.95] using simple random sampling. As a result, the historical inflow (which includes both dry and wet hydrological conditions) is decreased in the different scenarios by as much as 40% (as the product between the historical trajectories and the multipliers). Such methodological procedure for the generation of the states of the world is often referred to as the delta method (Brown et al., 2012), and has been widely used in the literature (see for example Bertoni et al. 2019). One of its main drawbacks is not being able to account for a seasonal shift in the hydrological regime which would naturally follow a change in the hydroclimatic patterns. However, the 110000 (10000 multiplier perturbations of the 11 years of historical data) hydrological years upon which the system is simulated provide a states of the world discretization grid which is dense enough to consider both the extremes and the intermediate scenarios over which the robustness of the various operating policies is computed.

2. **Irrigation demand uncertainty**: Droogers et al. (2014) estimated the expansion in irrigated area in the study site to be up to 25%. For the irrigation demand trajectories we, therefore, assume a constant irrigation multiplier, drawing $K$ samples from an uniform distribution over [1, 1.25] using simple random sampling. As a results, $K$ irrigation demand scenarios are generated as the product between each multiplier and the historical irrigation demand time series.

3. **Additional inflow from Corumana dam uncertainty**: The World Bank is currently financing the Greater Maputo Water supply Expansion project, which started in 2013 and is still under development. The project is compounded of a water treatment plant downstream the Corumana dam, and by a 95 km pipeline connecting Corumana with the city of Maputo, ensuring an additional water supply of $1.8 \, m^3 s^{-1}$. The completion day $c_d$ of the project is treated as a stochastic variable,

for which $K$ samples are drawn from a uniform discrete distribution in the interval [0, H], where H = eleven years (i.e., the simulation horizon for an individual SOW). As a result, $K$ additional inflow from Corumana scenarios are generated. In each scenario, the inflow from Corumana on a day $t$ is computed as:

$$q_t^C = \begin{cases} 1.8 & \text{if } t \geq c_d \\ 0 & \text{else} \end{cases} \tag{13}$$

4. **Urban water demand uncertainty** growth for the city of Maputo to be up to 2% per year. Urban demand multipliers are therefore assumed here to be constant every year, and a sample of size $K$ is extracted from an uniform distribution within the range [1, 1.02]. Therefore, the urban demand trajectory in a certain scenario can be computed as the exponential increment of the historical urban demand.

As a result, $K = 10000$ deeply uncertain SOWs embedding inflow, irrigation demand, additional inflow from Corumana and urban demand scenarios are generated. Eeach of the $K$ set could be seen as a vector composed by four parameters (i.e., one multiplier for each uncertainty source), sampled from their uniform distribution. The set $\Xi$ including all the states of the world where the operating policies are evaluated is generated with a Latin Hypercube Sampling (LHS) across the four dimension of the uncertainty sources, with a sample size $N_u = 5000$ (cardinality of $\Xi$).

Table 3 provides a summary of the descriptive statistics of the SOWs aggregated by uncertainty source, including the driest and wettest year, as well as the maximum and minimum water demand, and the maximum and minimum yearly inflow from Corumana.

**Table 3.** Statistical extremes of the SOWs, aggregated by uncertainty source

| | | |
|---|---|---|
| Streamflow $[Mm^3y^{-1}]$ | Current value | 220 |
| | Driest year | 60 |
| | Wettest year | 342 |
| Upstream irrigation $[Mm^3y^{-1}]$ | Current value | 22.8 |
| | Minimum demand | 22.8 |
| | Maximum demand | 28.5 |
| Downstream irrigation $[Mm^3y^{-1}]$ | Current value | 11.5 |
| | Minimum demand | 11.5 |
| | Maximum demand | 14.4 |
| Urban demand $[Mm^3y^{-1}]$ | Current value | 80 |
| | Minimum demand | 80 |
| | Maximum demand | 98 |
| Inflow from Corumana $[Mm^3y^{-1}]$ | Current value | 0 |
| | Minimum demand | 0 |
| | Maximum demand | 56 |

### 3.3.2 Robustness metric

To select the most robust alternative for each of the stakeholders, we used the $minimax$ robustness metric. The computation of the metric requires $N_u$ simulations, one for each SOW $\chi \in \Xi$.

The $minimax$ identifies, among the optimal control policies $\pi^*$ designed via EMPODPS, the most robust alternative $\pi^r \in \pi^*$ as the one attaining the best performance in the worst among the SOWs:

$$\pi^r = arg\min_{\pi^*} \left( \max_{\chi \in \Xi} J(\pi^*, \chi) \right) \tag{14}$$

This metric, usually associated with a high risk aversion attitude, selects the alternative assuming that the worst future conditions will be realized (Wald, 1950).

### 3.4 Sensitivity and Uncertainty

In many hydrological applications, sensitivity and uncertainty analysis are often closely related, to the point that certain UA techniques, as scenario discovery (SD) and factor mapping (FM) could be assumed as SA methods. Yet, while UA is used for quantifying the uncertainty in the output, SA is typically adopted to apportioning output uncertainty to the different uncertainty sources (or input factors) (Saltelli et al., 2008). Following this line of thought, one could use FM or scenario SD to determine which uncertain input combination might cause robust policies to perform poorly. Those techniques usually classify the (uncertain) input samples as *behavioural* or *non-behavioural*, depending on whether the response variable (in this application: the objective function) exhibits a certain pattern or not. In our study the distinction between *behavioural* and *non-behavioural* perturbations is performed through the GLUE uncertainty analysis, which has been indeed developed starting from some of the basic ideas of Regional SA and factor mapping; while the PAWN SA method Pianosi et al. (2016) was used to identify the major sources of vulnerability. Despite their many similarities, the two techniques often offer a valuable complement to each other, with SA providing valuable extra insights to UA on the identification of the most relevant uncertainty sources.

### 3.4.1 GLUE

We perform the quantification of the output variability in response to the four uncertainty sources considered in the paper by employing the GLUE method. In particular, GLUE allows for determining which SOW result in optimal robust policies that yield unacceptable results. The implementation of GLUE encompasses several steps, most of which are already included within either the robustness or the sensitivity analysis.

A brief summary is provided below:

1. **Generation of the States of the World**: as described above, SOWs are obtained in this study by near-random sampling of the perturbed time series of inflow, irrigation demand, infrastructure and population.

2. **Specification of the objective function**: this function is defined in this study as each of the four objectives which constitutes the four-dimensional objective function in Equation 11b.

3. **Definition of the threshold values for the behavioral system perturbation set**: here, the threshold value is defined for each operating policy as the 5th percentile of all the objective function realizations, after simulating the system for $N_u$ SOWs. The behavioral perturbations are defined as the multiplier values keeping the objective function below the threshold.

From a computational perspective the GLUE algorithm requires, for a certain robust operating policy, to re-simulate the system for each $\chi \in \Xi$ (i.e.: for each of the states of the world generated during robustness analysis). At every simulation, the value of the objective function is computed and stored. Upon completing this step, an empirical cumulative distribution function is fit to each of the four objective function's dimensions (i.e.: the operating objectives). Then, the algorithm checks whether a state of the world leads or not to an objective value below a certain threshold (which in this study is set as the 5th percentile of the objective CDF). In the former case, the SOW (and the system perturbation set the SOW is constituted by) is classified as behavioral, in the latter as non-behavioral. Considering the expected adverse effect which most of the uncertain future realization of the external system drivers might have on water availability and demands, the choice of the 5th percentile as a threshold value is representative of the stakeholder desire to stay as close as possible to the historical performance.

### 3.4.2 PAWN

To identify the main sources of vulnerability across robust policies, we use the PAWN sensitivity analysis (Pianosi and Wagener, 2015). PAWN is a distribution-based method, and its choice lies in its applicability to nonlinear models (Amaranto et al., 2020), and its independency from the type of output distributions (for example, symmetric, multimodal or highly skewed). In addition, several studies (Zadeh et al., 2017; Pianosi and Wagener, 2018) have shown the capacity of PAWN to provide stable results for relatively low sample sizes.

In PAWN, the sensitivity of the output $y$ (in this specific case, the objective function value) to variations of an input $x_i$ (climate, agriculture, infrastructure and population) is measured as the distance between the unconditional and conditional empirical cumulative distribution (CDF) of $y$. The distance between distributions is measured by the Kolmogorov-Smirnov statistics, computed as follows:

$$KS(x_i) = \max_y |F_y(y) - F_{y|x_i}(y)| \tag{15}$$

where $F_y(y)$ is the empirical unconditional distribution of $y$, and $F_{(y|x_i)}(y)$ is the empirical conditional distribution of $y$ when the $i-th$ input is kept fixed at the nominal value $x_i$. Considering the dependence of KS on the nominal value, the PAWN method considers KS statistics over a prescribed number of nomimal values and then computes the sensitivity index as follows:

$$S_i = \max_{x_i = \bar{x}_i^{(1)}, \ldots, \bar{x}_i^{(n_c)}} [KS(x_i)] \tag{16}$$

Where $\bar{x}_i^{(1)}, \ldots, \bar{x}_i^{(n_c)}$ are $n_c$ randomly sampled values for the fixed input $x_i$. By definition, all the $KS(x_i)$ values, and consequently the sensitivity indices $S_i$, vary in the range [0, 1]. The closer the unconditional distribution $F_y(y)$ is to the conditional ones $F_{(y|x_i)}(y)$, the smaller the $KS(x_i)$ values and, therefore, the smaller the sensitivity of $y$ to $x_i$, and vice versa.

Operatively, $F_y(y)$ is computed for a certain policy by iteratively simulating the system for $N_u$ SOWs, generating therefore $N_u$ realizations of each objective upon which the empirical CDFs are constructed (Figure 4). The computation of $F_{(y|x_i)}(y)$ re-

quires $N_c$ SOWs simulations for each of the $n_c$ nominal value of $x_i$ (as shown in Figure 4 for $x_i = inflow$). The unconditional empirical CDF for each uncertainty source is therefore estimated upon $(n_c \cdot N_c)$ system realizations.

Considering $M$ uncertainty sources, the total number of system evaluations required for implementing PAWN for a single policy is therefore $N_u + (M \cdot n_c \cdot N_c)$, where $N_u$ and $N_c$ are the number of SOWs used for the unconditional and conditional distributions, respectively, while $n_c$ is the number of nominal value for the uncertainty sources.

In this study, we fixed $N_u$, $N_c$ and $n_c$ to 5000, 2000 and 12. The decision is based on the observation of the uncertainty bounds in the sensitivity index obtained by 100 bootstraps of the input-output realizations upon which it is computed.

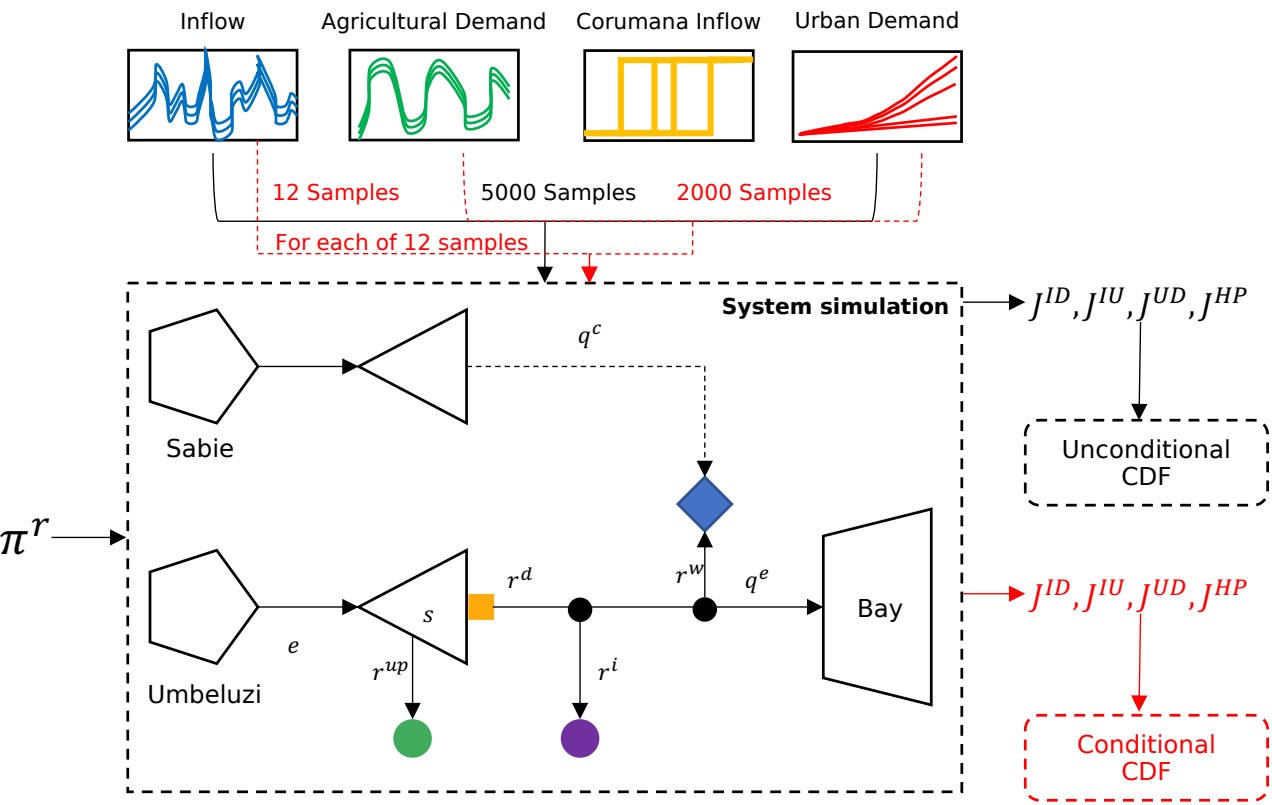

**Figure 4.** PAWN conceptual framework, unconditional objective function distribution in black and conditional objective function distribution in red.

## 4 Results and Discussion

### 4.1 Optimization: Multiobjective Tradeoffs

The Pareto-optimal policies obtained by solving the optimization problem defined in equation 11 are reported in Figure 5.
We run the optimization on the baseline: i.e. the historical hydrological ($\Delta$ inflow = 0), agricultural ($\Delta$ irrigated area = 0),
infrastructural (no external supply) and social (population growth = 0) conditions. Each line is a different policy, each axes
represents an optimization objective, and the crossing point identifies the objective value derived from the implementation of
a certain policy (normalized between minimum and maximum), and to be minimized. The ideal solution is a horizontal line
intersecting all four axes at their bottom. The extent of the conflicts is proportional to the slope of the lines connecting two
adjacent axis. The colour of the lines represent the most robust policy for each stakeholder, and will be discussed more in
details later.

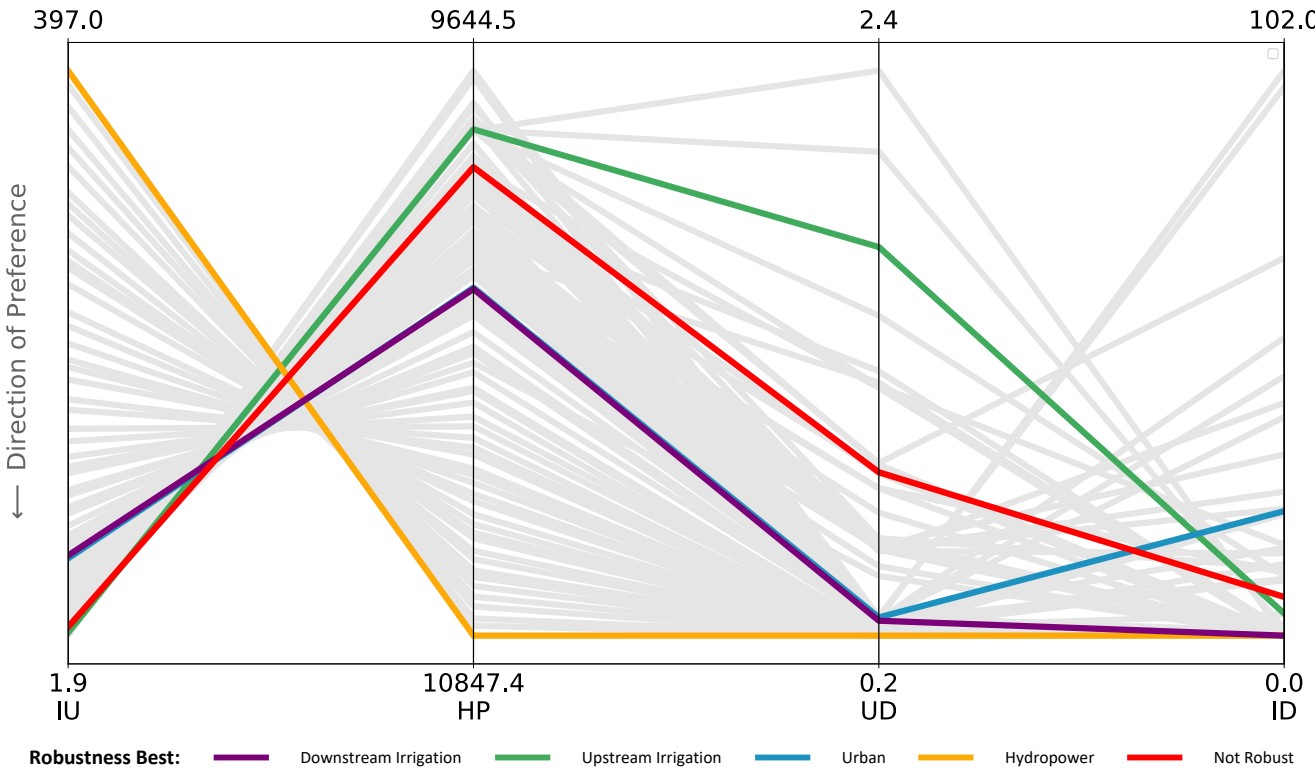

**Figure 5.** Policy performance across sectors under the baseline scenario. IU = upstream irrigation; HP = hydropower; UD = urban deficit, ID = irrigation downstream

.

Not surprisingly, the most conflicting objectives under the baseline scenarios are hydropower and upstream irrigation, since the latter is the only stakeholder that cannot benefit of the dam releases from hydropower generation, and subtracts a potential source of release for energy production by pumping out of the reservoir to satisfy crop requirements. The combined urban and irrigation demand downstream is about 50% the turbine maximum capacity in the power plant, and can therefore be satisfied by hydropower releases.

Compromise solutions between upstream irrigation, urban and downstream irrigation can be achieved at the expenses of hydropower (as can be seen by the purple line in Figure 5), with a 10% reduction in energy production leading to a near-zero deficit for all the other stakeholders.

## 4.2 Robustness: Probabilistic Tradeoffs

Figure 6a presents the most robust (MR) solutions for each stakeholder (defined as those performing best in the worst system configuration) under deeply uncertain scenarios, along with where such solution would fall when ranked for other stakeholders. The red line identifies a generic non robust (NR) policy, which generates the $90^{th}$ percentile in the overall robustness ranking.

Even though historically NR led to objective values close to those of the most robust solution for upstream irrigation (RIU, green lines in Figures 5 and 6), it experiences a dramatic performance degradation under deep uncertainty, highlighting the importance of robustness analysis in a multi-objective decision making framework for the discovery of optimal solutions with low vulnerability against the uncertainties in their forcing realization.

Furthermore, while in historical condition the robust solution for irrigation downstream (RID) and for the city of Maputo (RUD) produce substantially the same urban deficit (Figure 5), RID provides much higher objective value under deep uncertainty. Following the opposite line of thought, it is also evident how the historical conflicts between the two stakeholders are exacerbated under deep uncertainty, and RUD becomes one of the worst (94th out of 100 policies) to adopt for downstream irrigation. A possible explanation lies in the way the diversion dam is operated in water scarcity conditions (occurring mainly under deep uncertainty). RID, by fulfilling 99% of crop water demand, prioritizes irrigation and increases the urban deficit. RUD instead systematically ensures urban water supply to the city of Maputo generating irrigation deficit even in historical conditions.

The cumulative distributions of performance under deep uncertainty represented in Figure 6b, along with those found through multi-objective optimization (gray lines), confirm that: (1) NR policy, being on the right side of each box, systematically offers poor performances in all objectives; (2) deep uncertainty might generate disputes among downstream irrigation and the city of Maputo, with one of the highest irrigation deficits produced by RUD (Figure 6b1); and (3) the existing conflicts between upstream irrigation and hydropower are exacerbated by both water scarcity and increase in irrigation demand (Figure 6b2).

The almost straight vertical purple line at the top left corner in Figure 6b1 also evidences how downstream irrigation is overall the stakeholder less vulnerable to deep uncertain scenarios. This is also observable by the high density of grey policies close to the zero-deficit even in the worst scenario. In addition, about 80% of the optimal policies still ensure an irrigation deficit value within the historical deficit range (0, 102 $(m^3s^{-1})^2Y^{-1}$)

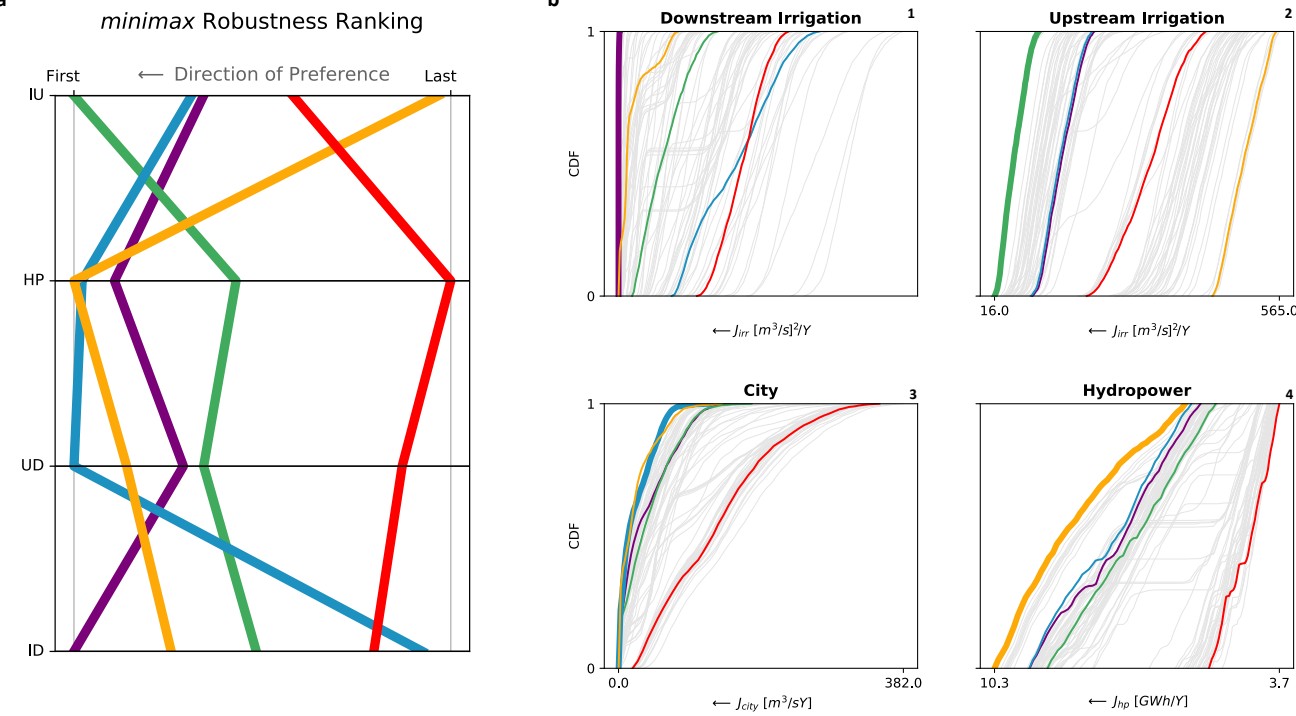

**Figure 6.** (a) Ranking the best solution according to each stakeholder, along with where these solutions fall when ranked according to other stakeholders. (b) Cumulative distributions functions for the four objectives using the most robust alternative per each stakeholder. The red line represent a non-robust solution.

By observing the cloud of gray CDF's, one could notice how some of the operating policies do not monotonically increase towards higher objective values, but instead shows sudden discontinuities. Such discontinuities are probably due to the realization of states of the world causing the reservoir level to reach the lower limit of the discretionary operating space. In other words, the variation in the external forcings (most likely a decrease in inflow) is such that, for certain NR operating policies, the reservoir level triggers a zero-release condition for a certain time-period. The absence of releases is reflected in the objective value, which suddenly increases due to no supply availability to cover the water demand.

Finally, the objective function of RHP shown in Figure 6b4 rapidly increases even for smaller system perturbations, making hydropower the most vulnerable sector to deep uncertainties. The possible explanation is twofold: the hydropower sector has the higher water consumption, and is solely dependent on inflow. Both conditions makes it particularly sensitive to adverse hydroclimatic scenarios, while the latter further exacerbate its sector vulnerability due to the absence of favourable infrastructural measures to mitigate water scarcity impact.

### 4.3 Sensitivity and Uncertainty

To allow a comparative discussion on how different SOW realizations unevenly shape the vulnerability of different control policies, we select the most and the least robust among the operational alternatives described above for each sector, and analyze the corresponding sensitivities and uncertainties. As mentioned before, hydropower solely depends on inflow, and exhibits the same qualitative behavior for all the operating policies. For this reason, results for this sector are not further discussed here.

Figures 7a and 7b represent a scatterplot of the downstream irrigation objective in correspondence with its relative forcing perturbation for RID (most robust alternative for downstream irrigation) and RUD (least robust alternative among those performing best for at least one stakeholder), respectively. Colours identify, for a specific policy, those SOWs whose realization causes the objective function to assume values below the fifth percentile (defined as the threshold values for the behavioral perturbation set). The robustness of RID against uncertain future condition is highlighted by the zero deficit produced in more than four thousands out of five thousands input realizations, and by the irrigation deficit below 1 $(m^3 s^{-1})^2 Y^{-1}$ in the remaining (Figure 7a). Furthermore, inflow represents the only influential factor for this control policy (as can be seen in Figure 7a1). In particular, irrigation deficit is produced only when the streamflow multiplier falls below 0.65. This practically means that, for any future climatic conditions generating streamflow reduction below to 35%, RID is able to ensure water supply no matter the expansion (among those embedded in the SOWs) in irrigated area.

Figures 7c and 7d show the value of the downstream irrigation sensitivity index (SI) to the uncertainty sources. The average value of the SI is represented by a black horizontal line, while the coloured box describes the 10-90 inter-quantile range derived by 100 bootstraps of the input-output realizations over which the SI is computed. The near-zero sensitivity values to agricultural, infrastructural and population uncertainties seem to confirm the above discussion on the robustness of RID (Figure 7c). When adopting RUD, instead, an irrigation deficit is generated as soon as any inflow perturbation occurs, and the objective function value (which increases up to 140 $(m^3 s^{-1})^2 Y^{-1}$) is also shaped by the expansion in irrigated area (Figure 7b), with sensitivity values to agricultural expansion up to 0.42 (Figure 7d). From a practical perspective, the main difference between RID and RUD stands in how an expansion in irrigated area might be perceived: as an opportunity in the first case, and as a potential source of conflicts in the latter.

As far as the urban deficit is concerned, the increase in the objective value during the 2010 drought (Figures 8a3 and 8b3) highlights the pivotal role of the Greater Maputo Water Supply Project for ensuring continuity of supply. However, when RUD is adopted, urban deficit stabilizes around 80 $m^3 s^{-1} Y^{-1}$ immediately after the event, and remains constant even if the project is not completed by the end of the simulation horizon. In other words, if the infrastructure is built before the drought, little or no deficit is generated. Otherwise, the system recovers from the event, and afterwards urban water demand is fulfilled no matter the construction time and the population growth. The opposite is true for NR, where deficit keeps growing, also augmented by the increase in water demand caused by population growth. This is also evident from the increased sensitivity to infrastructure and population, with SI values reaching values up to 0.83 and 0.32, respectively (Figure 8d). The comparison between RUD and NR suggests that the structural intervention in the former case is a fundamental action to undertake in order to cope with

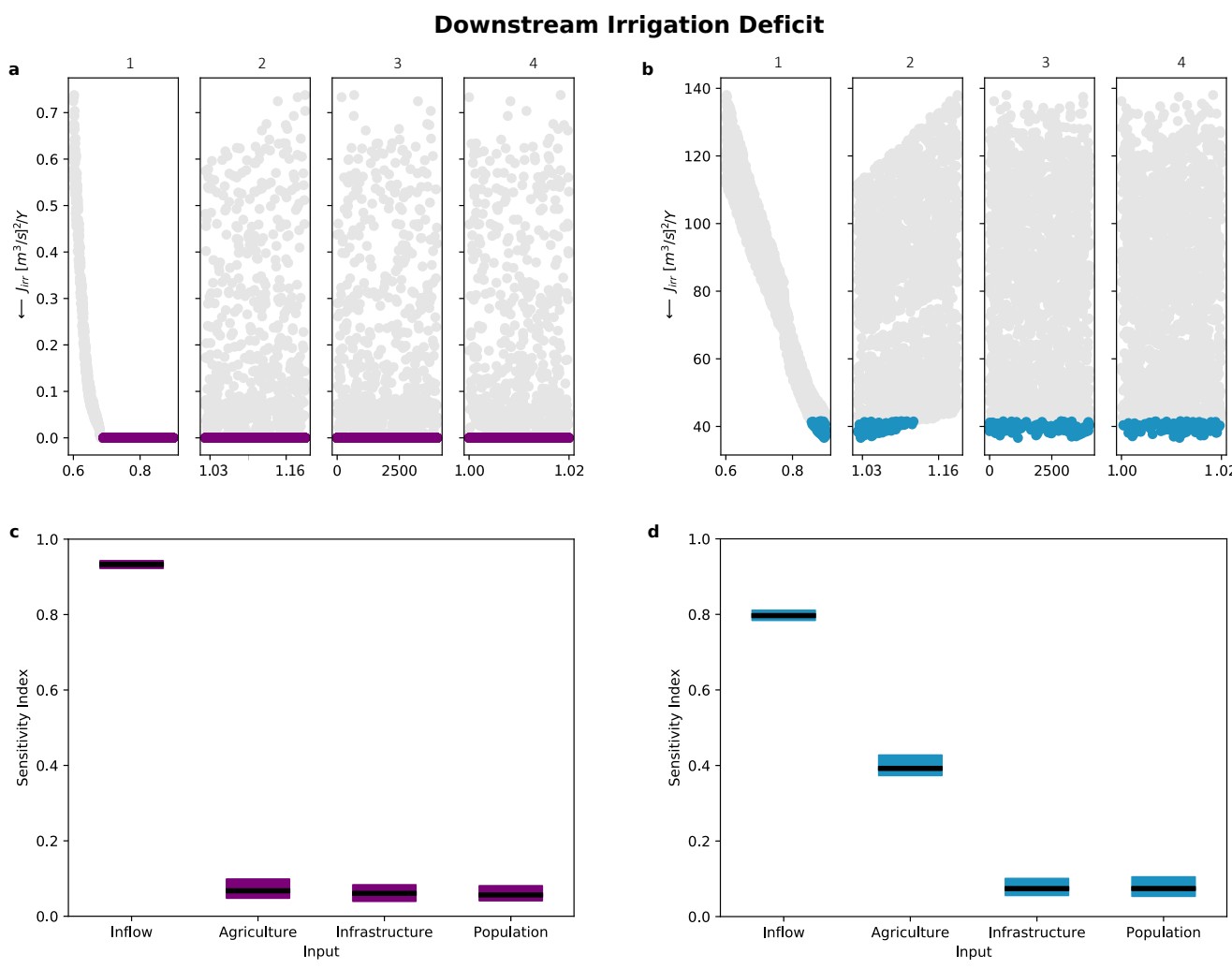

**Figure 7.** Behavioural perturbations (a-b) and sensitivity index (c-d) for the robust upstream irrigation (purple) and robust urban deficit (blue) policies.

extreme hydrological scenarios, while in the latter it becomes pillar for the everyday operation of the system, especially when population growth increases the water demand in the metropolitan area of Greater Maputo.

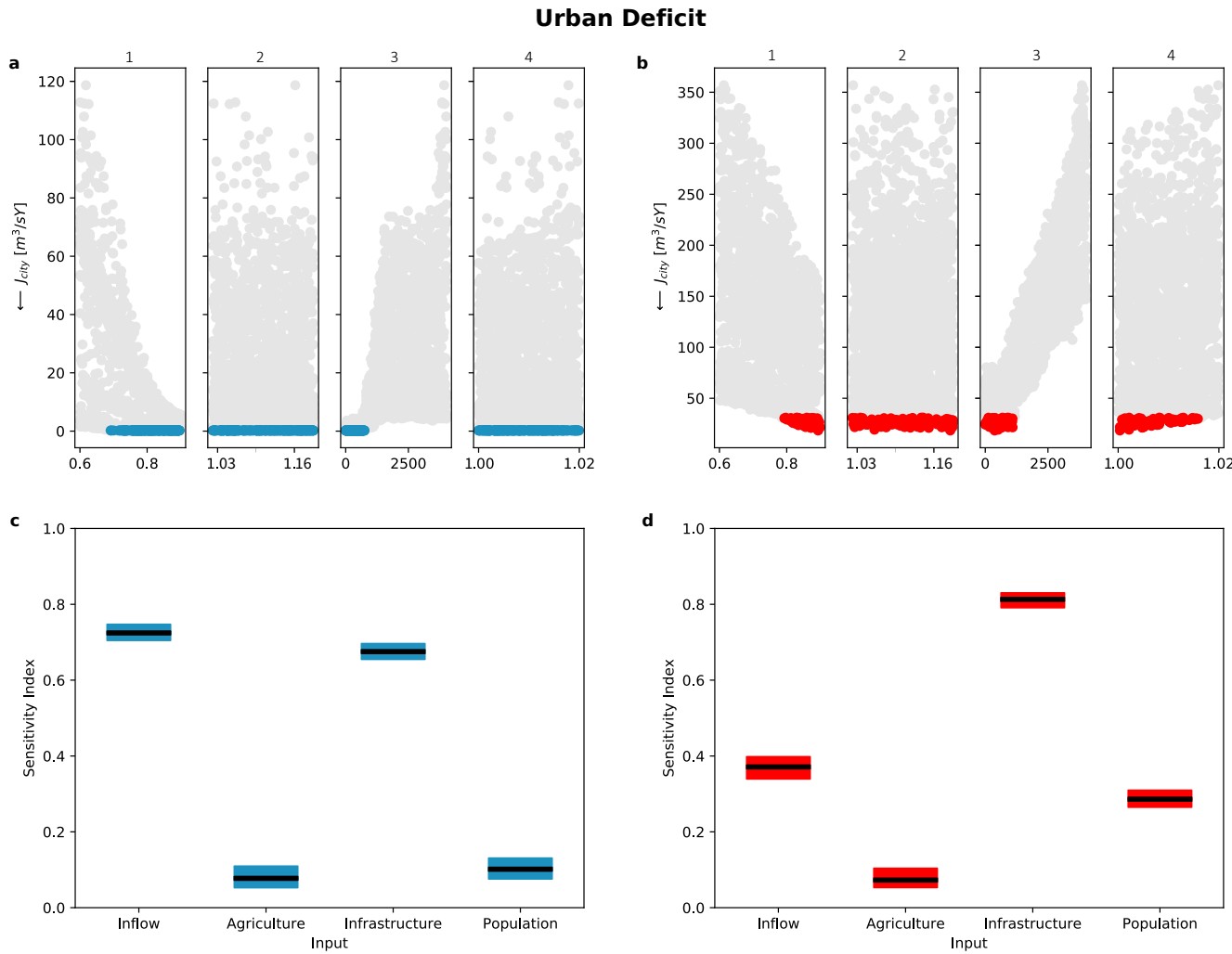

**Figure 8.** Behavioural perturbations (a-b) and sensitivity index (c-d) for the robust urban deficit (blue) and non robust (red) policies.

    Finally, upstream irrigation experiences a sudden increase in irrigation deficit as soon as inflow decreases and irrigation demand grows (Figures 9a and 9b). Unlike downstream irrigation districts, the upstream agricultural sector, not being able to 're-use' water from other sectors, is less flexible to a change in the external drivers, and therefore less robust to their uncertain

realizations. However, when comparing RUI (Figure 9a) and RHP (Figure 9b), it is evident that irrigation deficit grows unevenly among policies: it is dramatically high (over 500 $(m^3s^{-1})^2Y^{-1}$) in the latter, and substantially lower in the former.

The higher sensitivity (SI > 0.9) of upstream irrigation deficit to agricultural expansion occurring when RHP is adopted (Figure 9d) seems to identify a systematic pattern in the comparison among most and least robust policies. For all the stakeholders analyzed, the latter have not just been consistently vulnerable to inflow decrease, but also to the other non-hydroclimatic system perturbations. In other words: (1) infrastructural interventions become a must, as reflected by the increased urban deficit when the GMWSEP is not built (Figure 8b3); (2) population growth exerts a non-negligible pressure on the water system, and contributes towards increasing urban deficit (Figure 8d); and (3) agricultural expansion is consistently limited by lack of water availability (Figures 9b and 9d). The opposite is true for RP: in spite of being (as expected) vulnerable to a decrease in inflow (even if lower in magnitude with respect to NR policies), they rely on new infrastructure only in emergency conditions, and are able to sustain increases both in terms of population and irrigated area.

## 5   Conclusions

In this study, we implement an integrated decision-analytic framework combining optimization, robustness, sensitivity and uncertainty analysis to better understand the major sources of uncertainty for water supply strategies in the lower Umbeluzi river, Mozambique. Results provide important insights on the robustness and vulnerability of reservoir operation to exogenous perturbations in managing multiple, conflicting objectives.

In particular, the main findings of this paper are:

– Optimal reservoir operating policies exploring similar tradeoffs in current conditions might lead to substantially different results under deeply uncertain scenarios. Specifically, the non-robust optimal solution presented in this study was largely dominated by 90% of the operating policies across all objectives once the system is perturbed.

– In water scarcity conditions, some new (i.e. non detected under historical conditions) tradeoffs between downstream irrigation and the urban supply suddenly emerge. In fact, while it is possible to explore operating policies ensuring maximum satisfaction for both stakeholders in current conditions, the two stakeholders are found in systematic competition, with the most robust policy for Greater Maputo being in the 95th percentile (96th out of hundred) when ranked for downstream irrigation.

– Overall, downstream irrigation appeared to be the least vulnerable stakeholder, with about 60% of control policies ensuring an objective value in the worst possible condition lower than the maximum computed by forcing the system with the historical trajectories.

– Robust policy for downstream irrigation ensures sustaining agricultural production with near-zero deficit for the majority of the agricultural water demand scenarios considered in this study. The only source of vulnerability in this case would be a streamflow reduction, which however produces only a marginal deficit increase even in the worst condition. The opposite holds true when choosing the less robust policy, with objective values suddenly increasing even for small perturbations in streamflow and irrigation demand.

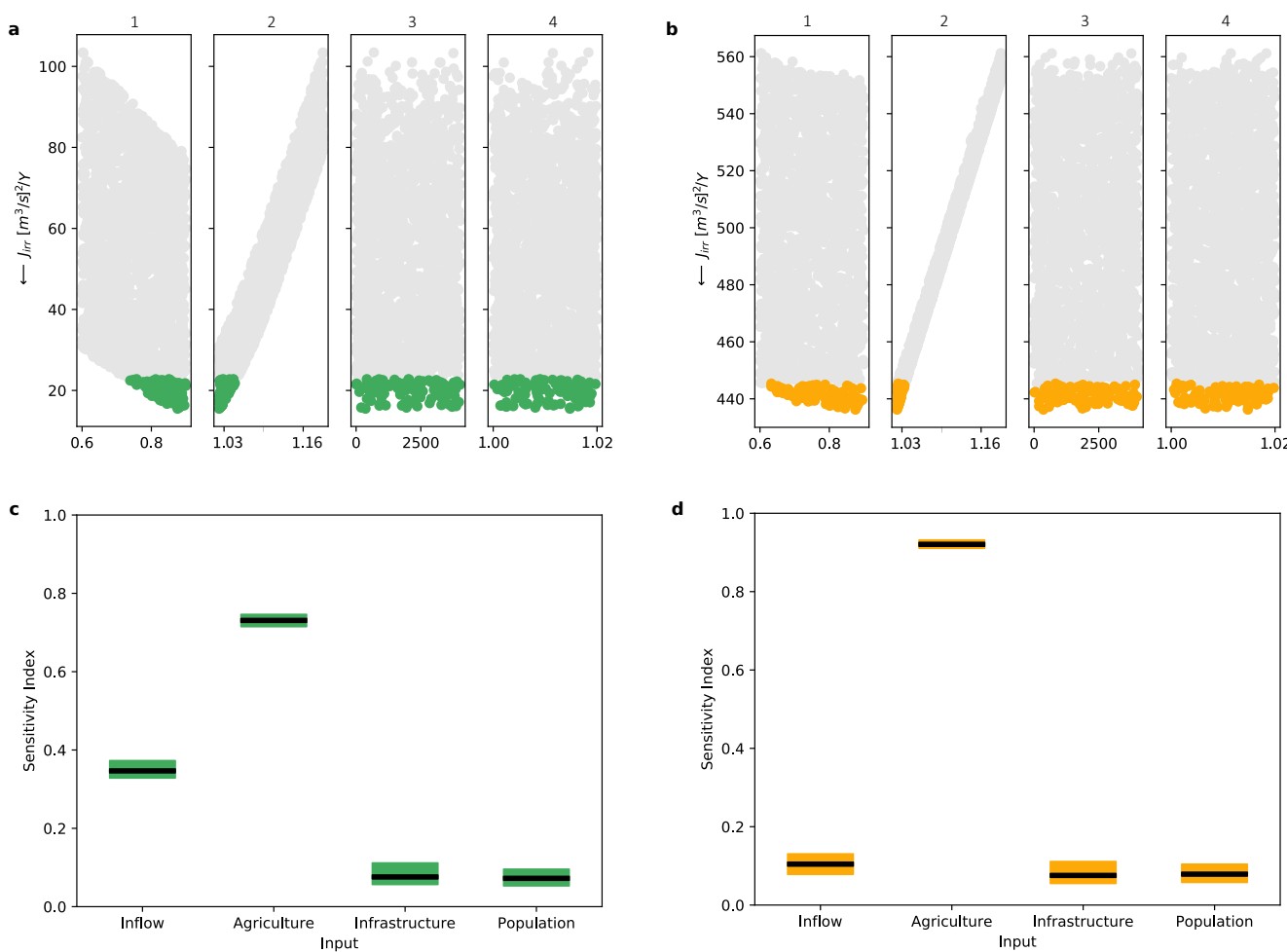

**Figure 9.** Behavioural perturbations (a-b) and sensitivity index (c-d) for the robust upstream irrigation (green) and robust hydropower (yellow) policies.

– As expected, hydropower production resulted to be solely dependent on inflow realizations. Furthermore, it was the least robust sector among those considered, with fast decrease in objective function values as soon as the system is perturbed.

– The implementation of the greater Maputo water supply expansion project appears to be vital for sustaining urban water supply. However, while its role can be envisioned solely as 'drought mitigator' when adopting robust policies (even for maximum population growth rates), it becomes essential to mitigate day to day deficits for non-robust solutions.

– Overall, it is possible to conclude that robust policies are usually vulnerable only to hydrological perturbations and are able to sustain the majority of population growth and agricultural expansion scenarios. Moreover, infrastructural

interventions become crucial only in extreme drought conditions. On the contrary, non-robust policies are sensitive also to social and agricultural changes, and require structural interventions to ensure stable supply.

From a methodological and computational perspective, the proposed decision analytic framework could be easily applied to any type of water system. However, in this study it is customized for the BPL river basin, and so are the conclusions and discussions thereof. Therefore, a possible future research path could be directed towards testing the proposed methodology across a range of hydro systems, in order to assess how stakeholder's tradeoffs, policy robustness and vulnerabilities are shaped by inherently different water availability and demand trajectories. The aforementioned results in terms of UA and SA are specifically tailored upon the parametric input perturbation set employed to generate the SOWs. Even though each perturbation set can be well documented from the literature, unexpected changes in one of the exogenous factor (i.e., higher population growth or lower streamflow availability) could shape the behavior of the system, altering therefore the extent to which a certain policy is vulnerable to each uncertainty source. One of the main limitations of this study is the assumption of independence among the distributions used for the generation of the historical trajectories multipliers which constitute the foundation for developing the states of the world. Even though such assumption allows to explore the full range of variability of the exogenous drivers of the system, and has been successfully applied in recent hydrological applications (see for example Pianosi and Wagener 2016 and Amaranto et al. 2020), a study entangling the covariance among the uncertainty sources could provide further insights on the robustness of each operating policy, and can tailor the sensitivity analysis on a more reliable set of perturbations. It is therefore a recommendation for a future research exercise. In this study, the sources of vulnerability and the uncertain output realizations of the optimal water supply strategies are investigated only for robust policies. Therefore, RA and SA are sequential methodological steps rather than being interdependent. Future research could be developed towards implementing SA methods that establish feedback loops with RA, in order to enhance the robustness of operating policies against uncertain exogenous factors. Furthermore, robust policies are selected here according to the *minimax* robustness metric. *minimax*, by identifying the best alternative in the worst possible input realization, encompasses a risk-adverse behavior of the decision maker. However, other criteria are available in the literature (see Giudici et al. 2020 for a review), each of them representing a different level of risk perception and its associated definition of robust operating policy. For a more comprehensive analysis, one could include additional robustness metrics in the experimental setup, and evaluate how the choice of the metric shapes the selection of robust policy and the identification of the sources of vulnerability thereof. A possible architecture in this regard can be found in Herman et al. (2015), where four different robustness definition are considered, and robust planning decisions are developed accordingly.

*Data availability*. The hydrological variables were provided by the Administracão Regional de Ãgua (ARA) Sul, but they are not made publicly accessible due to provider's request. Any request for access to the data can be addressed to Alessandro Amaranto (alessandro.amaranto@polimi.it). Temperature and Precipitation projections were obtained by the Earth System Grid Federation (ESFG, https://esg-dn1.nsc.liu.se/search/cordex/), while the numerical experiment developed for their downscaling is available at https://github.com/mxgiuliani00/

ClimateScenarioAnalysisToolbox/tree/master/QuantileMapping/gridded_data. Downscaling realizations, as well as the experimental results as described in this paper are available at https://github.com/alessandroamaranto/Moz_SA (folder in preparation).

*Author contributions.* Conceptualization: AA and AC; Data curation AA and DJ; Methodology AA and AC; Investigation AA and AC; Original draft AA; Writing, reviewing and Editing AA, AC and DJ

*Competing interests.* The authors declare that they have no conflict of interest

*Acknowledgements.* The authors gratefully thanks the Administracão Regional de Āgua Sul for providing the observed hydrological data. This research was supported by Politecnico di Milano within the project BOA_MA_NHÃ, Maputo!. Finally, the authors thank Davide Danilo Chiarelli and Maria Cristina Rulli for providing irrigation demand data.

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
