# Peer review of "Disentangling Sources of Future Uncertainties for Water Management in Sub-Saharan River Basins"

_Hydrology and Earth System Sciences, 2021_

## Referee Comment (RC2)

Review of "Disentangling Sources of Future Uncertainties for Water Management in Sub-Saharan River Basins", Amaranto et al.

This paper presents a study that applies a decision framework for developing optimal operating policies for the Barragem de Pequeños Libombos (BPL) Reservoir in Mozambique. This reservoir has been constructed to serve the water supply to the greater Maputo area. It also serves irrigation up and downstream of the reservoir, and a hydropower station to generate energy. A multi-objective evolutionary optimisation approach is applied to find optimal operating policies. The robustness of these policies to changes in irrigation demand, urban demand, and climatic conditions is then explored. Finally, the sensitivity of both robust and non-robust policies to these changes is explored, concluding that robust policies are sensitive primarily to changes in climatic conditions and consequent changes in reservoir inflows, while non-robust policies are also sensitive to population growth and structural interventions. This last points is perhaps one of the more interesting aspect of the contribution to the readership of HESS. Besides that the paper seems to focus primarily on the application of the optimisation approach and exploration of robustness and sensitivity. In this sense the scope of the paper appear to hinge on two thoughts, one being the optimisation approach that is proposed and in particular the exploring the sensitivity of optimal polices, and the second the application to the case of the BPL reservoir.

Though the application of the methodology is of interest, and to some extent of an interest to the readership of HESS, the manuscript does not reflect on the findings in the broader context than the particular case study in which the method is applied. As a result it is also not entirely clear to what extent the conclusions established are case study specific, or if these can be made more generically.

This would require a much improved discussion interpretation of the results. This would also be more appealing to the scope of HESS if that discussion is presented within the hydro-sociological context and how the specific choices and assumptions that are made reflect on the generality of the conclusions. In the general comments below this is elaborated on.

Overall the paper is well written and well structured. However, the authors have not really paid much attention to the correct use of grammar. While a possible copy-editing phase would easily solve several issues, a more careful preparation would have been expected. There are many sentences with multiple grammatical errors. Some examples are included in the detailed comments, though these are not considered to be exhaustive but rather as illustrative.

**General Comments**

It would be very helpful if more details are provided on the hydrological characteristics of water availability and demand, and also reflecting on the size of the reservoir in relation to that demand. First, it would be useful to be explicit as to the average availability of water. A rough estimate made from figure 2 would suggest that the mean annual inflow is about 7.5 $m^3$/s, with 10-90 percentile variability as indicated. This constitutes about 236 $Mm^3$ per year. The volume of the reservoir is reported to be 382 $Mm^3$ per year. This would then be 50% more than the average annual inflow, which means that for this reservoir the inter-annual variability is of importance. The paper does mention that a part of the storage is inactive, which also influences the sensitivity of the policies, but it is not clear how much this inactive storage is. Reviewing the numbers pertaining to demand, the irrigation demand (upstream and downstream) is mentioned to be some 33 $Mm^3$ per year on

average. The water supply demand is stated to be 80 $Mm^3$ so it is clearly the dominant demand, which is in line with the purpose of the reservoir (it would be useful to understand the date for which this demand has been established). Demand to meet environmental needs is set at 15%, with a simple estimate being about 35 $Mm^3$. In any case, these numbers suggest that total demand (irrigation + urban + environmental) used as a benchmark is about 62% of the average annual inflow. It is also some 38% of the storage of the reservoir. Though these percentages have been estimated somewhat roughly based on the data, they are useful to interpret the results, to understand the degree of water stress, and also how well the reservoir can buffer the variability of the inflow to meet demands. I believe the authors should develop a more extensive reflection on how the results they find depend on these ratios of availability and demand, as a more stressed situation where availability and demand could well yield quite different results. It would also be important to understand the inter and intra annual variability of the inflow, related to these ratios, and a more elaborate discussion of how these ratios influence the conclusions on robustness and sensitivity. What would happen if, for example the available storage of the reservoir was much smaller than the average annual inflow, or the average annual demand? The numbers presented as a baseline do not suggest a situation that is under extreme stress. Also, the changes to irrigation demand (25%) and Urban demand (2%) constitute in absolute values only some 8 $Mm^3$ and 1.6 $Mm^3$ annually. This appears to be somewhat modest in relation the average annual inflow and the size of the storage, which would suggest variability and uncertainty can be easily buffered. I would be very curious if the authors could reflect on how these values influence their conclusions on the importance of the variability of the hydrological inflows which suggest a reduction of between 5% and 40% of the inflow, constituting some 12 to 95 $Mm^3$, so already much more dominant from the outset. The manuscript would increase in scientific interest if a more elaborate discussion is provided that reflects on the ratios between the available water and the demands, as well as the relative magnitude of variability and changes.

I have some doubts on the formulation of the optimisation problem, and how representative this is for the case being studied. It appears that the water supply to Greater Maputo is considered a lower priority to meeting irrigation demand. There is a hedging rule applied in Eqn 3, but ultimately the water supply of Maputo is established as the remainder of the actual downstream release, after actual delivery to irrigation and meeting the environmental flow demand (eqn.5). In the introduction, however, it is noted that the reservoir is currently operated in a different way. The supply to Maputo gains priority over irrigation, with the latter being stopped if there is insufficient water. This suggests a fundamentally different policy is followed than that which is suggested here. This raises the question how realistic the proposed approach in modelling the operating policy is, and what the impact of this limitation is on the findings and conclusions. I would expect that a much higher sensitivity to population growth would then be found, and that the competition between urban supply and irrigation would be exacerbated, as in more extreme growth population growth scenarios this prioritisation would mean irrigations may become unfeasible. I would again think the manuscript would gain in interest if a discussion is provided as to the extent to which the formulation of the optimisation problem reflects the actual operation of the reservoir, as well as how that formulation influences the conclusions found.

Related to this comment is also the question on how realistic the scenarios chosen are. The growth rate of Maputo has been set at some 2%, which seems modest. Those familiar with the area would know that Matola is growing much faster than Maputo, at some 4% per year, and is already larger than Maputo itself. Also, can the growth in demand be considered as linear with population? Several

studies show that increasing development also result in an increase in water demand disproportional to population growth. Given the dominance of the urban demand, and the very modest increase such factors could well be discussed. It may be that the conclusions found on the sensitivity of robust strategies are indeed valid within the very modest growth scenarios explored. There may well be a limit to the change in which the conclusion on sensitivity holds, after which this sensitivity may increases dramatically. This would help generalise the findings of the paper.

Continuing on from this comment, it would also be useful to understand the rationale of the objective functions themselves. Hydropower is clearly a by-product in the operation of the dam as there is no demand to meet (or a penalty if such demand is not met) and simply maximises the profit from generation. Given the primary purpose of the dam that would appear reasonable. The other objective functions, however, apply a squared residual to irrigation deficits and a linear residual to urban water supply deficit. This seems somewhat incongruous to already noted priority of urban water supply to irrigation, as it would tend to minimise irrigation deficits with respect to deficits in urban supply. I agree that the factors in eqn.3 influence this preference, but it would be good to understand the rationale behind this choice. A more detailed description of the water allocation policy to irrigation would be useful. If there is insufficient water, and given that the crops are predominantly cash-crops, it would make sense that farmers simply scale the cropped area - which would suggest a linear loss if demands are not met.

Further discussion should also be provided on the scenarios developed to represent climate change. The method chosen is simple, which I agree is appropriate within the context of this study. However, if I understand correctly the change is a linear factor applied to the whole time series. This does not resolve any changes to the inter-annual variability of inflows. I am not fully familiar with climate outlooks for this particular basin, but presume the influence of climate change may be more multi-dimensional than a simple proportional reduction across the year. There could also be changes in the distribution of flow across the wet and dry season. Such shifts are of important to reservoir operation policies. This may be relevant to the ability of the reservoir to the meeting of irrigation demand, which is (logically) primarily in the mid and late dry season.

**Detailed Comments:**

Please revise the grammar in the whole manuscript. Here is an example of one or two sentences with multiple grammatical errors. All minor and easily resolved, but a careful grammatical review could improve the readability (green indicates and addition, red-strikethrough a deletion, while text in blue indicates where the text does not make sense, grammatically).

Lines 33-38

An archetypal example of a highly regulated, fast evolving South-Saharan hydrosystem is the Lower Umbeluzi river basin, Mozambique. About 45 km upstream of its delta in the Maputo bay, the river flows in the Barragem de Pequenos Libompos reservoir, which is operated to balance hydropower production, urban supply to the two million inhabitants of the Maputo province, and irrigation supply for to the 3600 ha of agricultural districts, mostly growing tropical fruits and sugarcane. A currently undergoing five year long drought have has boosted crop prices by about 50%, hindering food access to a population currently growing at rate of 0.6% per year and exacerbating conflicts among the urban, agricultural and energy sectors.

currently undergoing: this is not grammatically correct as it is not clear what is currently undergoing a five year long drought.

Line 38: Fryslân et al. 2014 is a very peculiar reference. In fact Fryslan is the organisation that published the report and is not a person. Either the organisation should be used, or the actual authors – but not a mix. The details of the reference in the reference list are insufficient and need to be completed.

Line 42: The world bank is funding

Lines 44-47: This section describes that the policy makes have certain needs. How were these needs ascertained? Was this a discussion held with these stakeholders, or are the fact that these are needs based on an assumption made by the authors?

Line 51: I am not sure this paper can be said to be tackling the evolution of the system drivers. Perhaps *identifying* would be a better word.

Line 71: *what* does not seem the correct word – please rephrase.

Lines 80-88. Please provide more detail on the dam and its outlets. In particular the size of the inactive zone (see general comments) and the maximum capacity of the penstock, and if the penstock is the only controlled outlet structure. This is suggested to have a capacity of 2x the combined downstream irrigation and urban demand, which would be 2.9 m3/s. Back-calculating with an installed capacity of 1.8 MW this would suggest the turbine efficiency is 73%, which is quite low. Please provide the correct numbers to help interpret the various results. In several places in the text it is suggested these are known, but they do not appear to be explicitly mentioned. Additional discussion should also be included on the presence of upstream reservoirs in the basin in Eswatini. These do exist.

Line 80: upstream of the estuary

Line 81: goal of supplying

Line 82: in Line 36 it is mentioned that irrigation is used for tropical fruits and sugar-cane. Here bananas as mentioned (which are tropical fruits). What happened to sugar cane?

Figure 2a. I do not fully understand the red line in this figure, but assume this is the median daily inflow. However, there are several peaks where the red line is at the limits of what the caption suggests is the 10-90 percentile range. Also, during the dry season, there seems to be an almost constant maximum to the 10-90 percentile range. Please clarify these data as these are if I understand correctly simulated data from the HBV model.

Figure 1a. The upstream irrigation areas in the map are incorrectly coloured. Also, the figures show there are unregulated tributaries. To what extent do these have a role in meeting demand?

Line 106: I assume the water is pumped to the irrigation districts and not from.

Line 129: Check spelling

Line 177/179: Please clarify the units for hydropower. Also, please provide the value of the efficiency of HP production applied.

Line 199: the sentence "which are known for well-validating out of sample data" needs to be rephrased as it is not clear

Line 234-260: The description of the samples is not clear. It would be clearer to describe each set of samples as a vector that is made up of the four parameters that sampled from their respective (uniform) distributions. Then K-vectors are sampled.

Line 254: What is value of the project horizon H?

Line 286: Please be specific on the likelihood function used. Is this constructed based on the simple summation of the four constituent objective functions? Please clarify how these are weighted.

Line 289: What is the rationale of selecting the 5%? Is there any relationship to impacts? Also it is not completely clear if this is the 5th exceedance of 5th non-exceedance percentile. Please clarify.

Line 371-374: It is somewhat confusing to refer to hydropower having a demand in the context here. Hydropower is generated as a by-product of the release to downstream irrigation and to meet urban demand. I would rather rephrase it as hydropower production.

Line 378-379: The grammar of this sentence is very poor. Please improve.

Line 387-388: Could the ability to meet the RID to endure water supply no matter the expansion of the irrigation area be linked to the size of the area. The maximum change of the d/s irrigation is on the order of 2.7 $Mm^3$ which is quite a limited amount (~3.4%) compared to the benchmark public water supply (see also general comments)

Line 394: Please check the units (also in other lines). What does 140 $m^3$/s/Year mean? This would suggest this is the mean annual flow? That does not appear to be correct. Please check this and other similar units.

---

## Author Response (AR1)

*Editor's comment*

**Reviewers' comments were quite useful, and the authors provided extensive answers and clarifications (and also rebuttal to some points), making it clear how the manuscript will be revised. The paper can move to the next stage, so I invite the authors to work on the revision.**

We would like to thank the two reviewers and the associate editor for their input and suggestions that we believe helped us to improve the quality of the manuscript further.

Please, find here below our response to the reviewers' comments in which RC stands for *'Reviewer Comment'* and AR stands for '*Author Response*'.

For simplicity, reviewers suggestions regarding grammar and typos were implemented directly in the manuscript.

In addition, we have included the new text in the updated version of the manuscript using blue colour, and the deleted text from the previous version in .

**Reviewer 1**

**RC:** *I enjoyed reading the paper, it is well written and it addresses both a methodological (i.e., integration of UA and SA into decision-robustness frameworks) and practical (i.e., the management of Umbeluzi river basin) issues. I support the effort in integrating UA and SA into established decision-support frameworks, I do however have the following issues:*

**AR:** We thank the reviewer for this outstanding review. His constructive comments helped us improve the manuscript and strengthen our analysis.

**General comments**

**RC:** *one of these frameworks proposed in the literature (perhaps the first one), i.e., Robust-Decison Making (RDM), in fact finds robust solutions heavily relying on a technique therein called Scenario Discovery, which to me can rightfully be assumed as a SA technique (i.e., factor mapping as also pointed out by Pianosi et al. (2016) http://dx.doi.org/10.1016/j.envsoft.2016.02.008). RDM has also been extensively adopted to address water planning problems https://link.springer.com/chapter/10.1007/978-3-030-05252-2_7. My suggestion would be to integrate such literature in the intro where it is stated that only few studies dealt with this problem before.*

**AR:** We thank the reviewer for the comment. Sensitivity Analysis has indeed been previously applied in robust decision making, and the suggested literature will be added into the manuscript. However, it might also be worth to point out few key aspects:

1. The reviewer is right: factor mapping (FM) and scenario discovery (SD) are key to determine which uncertain input combination might cause robust policies to perform poorly. Those techniques usually classify the (uncertain) input samples as 'behavioural' or 'non-behavioural', depending on whether the response variable (in this application: the objective function) exhibits a certain pattern or not. Even though both FM and SD can both be assumed as SA techniques, in our study the classification between "behavioral" and "non-behavioral" perturbations is performed through the GLUE uncertainty analysis (Beven and Freer, 2001). Nonetheless, as mentioned by Saltelli et al., (2008) and Pianosi et al., (2016), UA and SA are closely related; to the point that their main distinction can be considered that '*UA focuses on quantifying the uncertainty in the output of the model, while GSA focuses on apportioning output uncertainty to the different sources of uncertainty*'. For instance, the GLUE UA has been developed starting from some of the basic ideas of Regional SA and factor mapping. Following the above, SA is not used here for scenario discovery, but builds upon it by providing a sensitivity index which measures quantitatively the main sources of vulnerability for reservoir operation.

We have clarified these aspects in the revised manuscript in section 3.4.1 as follows:

> In many hydrological applications, sensitivity and uncertainty analysis are often closely related to the point that certain UA techniques, as scenario discovery (SD) and factor mapping (FM) could be assumed as SA methods. Yet, while  UA is used for quantifying the uncertainty in the output ,  SA is typically adopted to apportioning output uncertainty to the different uncertainty sources (or input factors) (Saltelli et al., 2008). Following this line of thought, one could use FM or SD to determine which uncertain input combination might cause robust policies to perform poorly. Those techniques usually classify the (uncertain) input samples as behavioural or non-behavioural, depending on whether the response variable (in this application: the objective function) exhibits a certain pattern or not. In our study the distinction between behavioural and non-behavioural perturbations is performed through the GLUE uncertainty

> analysis, which has been indeed developed starting from some of the basic ideas of Regional SA and factor mapping; while the PAWN SA method (Pianosi et al., 2015) was used to identify the major sources of vulnerability. Despite their many similarities, the two techniques often offer a valuable complement to each other, with SA providing valuable extra insights to UA on the identification of the most relevant uncertainty sources (Pianosi et al., 2016).

RDM in water resources has indeed been applied to address water planning problems. However, fewer are the studies where quantitative SA (i.e. where each input factor is associated with a quantitative and reproducible evaluation of its relative influence) has been employed for reservoir operation in a multi-objective context.

To account for the important application of RDM to pure planning problems, the literature proposed by the reviewer have been integrated in the manuscript as follows.

> So far. even though it has been recognized that optimal planning and control methods should employ SA to identify water resources system vulnerabilities to both structural and parametric uncertainties (Herman et al., 2019), only few studies developed quantitative analyses tosupport water resource planning (e.g. Herman et al., 2015; Trindade et al., 2017, 2019; Groves et al., 2019).

*Beven, K., & Freer, J. (2001). Equifinality, data assimilation, and uncertainty estimation in mechanistic modelling of complex environmental systems using the GLUE methodology. Journal of hydrology, 249(1-4), 11-29.*

*Pianosi, F., Beven, K., Freer, J., Hall, J. W., Rougier, J., Stephenson, D. B., & Wagener, T. (2016). Sensitivity analysis of environmental models: A systematic review with practical workflow. Environmental Modelling & Software, 79, 214-232.*

*Saltelli, A., Ratto, M., Andres, T., Campolongo, F., Cariboni, J., Gatelli, D., ... & Tarantola, S. (2008). Global sensitivity analysis: the primer. John Wiley & Sons.*

**RC:** *perhaps related with the previous point: I think the UA and SA steps should more strongly be integrated into the robustness search. Afterall, robustness broadly defines the performance of the system under uncertainty and the UA and SA steps should be* part *of it. In the proposed framework instead (Fig. 3), the search for robustness and the UA and SA steps are reported as two different steps. How can the intial robust policies be updated/ameliorated based on SA analysis results? I think one could change the current framework either bringing the UA/SA step into robustness, or establishing some sort of feedback loop between the two. This would also affect the results, where it would be great to comment upon how, in light of the GLUE and PAWN results, one could increase the robustness of the previosuly found strategies. This could perhaps come in the form of a table where e.g. a description of how the best policies for each stakeholder change going through the three steps of the frameworks, i.e., from optimal to robust to robust+SA.*

**AR:** We thank the reviewer for the comment. However, robustness, uncertainty and sensitivity analysis are already deeply interconnected into the proposed framework.

Such interconnection can be observed from three different perspectives:

1. RA, UA, and SA are all based upon similar mathematical techniques, since they all rely upon simulating iteratively the system following a perturbation on the input set.

2. RA, UA, and SA are all performed using same set of input perturbations, i.e., the very own states of the world generated during robustness analysis are also those employed during both UA and SA. The idea behind this choice is that, given a set of scenarios, robustness analysis identifies the most robust policies, UA determines those scenarios leading to acceptable system performances and SA provide a quantitative measure of the vulnerability of each sector to changes in each uncertain input factor. This consideration leads to:

3. During robustness analysis, the states of the world are used to force the model in order to discover robust policies. However, robustness analysis per se does not ensure any knowledge of the scenarios which might yield to acceptable system performance (behavioral perturbations) or any insights upon a quantitative measure of the specific sensitivity (i.e., sensitivity index) of each dimension of the objective function to the various input realizations. Conversely, while UA and SA provide knowledge on each of the aforementioned aspects, they are not performed on the whole set of alternatives designed via optimization, but only on those who are robust (discovered via robustness analysis) against the system perturbations embedded in the states of the world. Considering this, they offer a complementary perspective about how a certain policy behave (i.e.: is it robust? What is the range of variability of the objective function? Which factor is most likely to yield to a failure in meeting a specific objective?) in response to the perturbations themselves.

In addition to the above, the quantitative measure of vulnerability offered by the sensitivity index is specifically tailored upon the policy for which it is computed for. Therefore, it is to be intended as valid only for the policy itself, with little or no possible extrapolation in terms of its robustness. For instance, let's suppose to have two policies (A and B). Let A to be characterized by marginal variations in the objective function value for all the system perturbations (i.e., robust), while the opposite is true for B. If, among all the input factors, streamflow is the responsible for most of the output variability for both A and B (no matter the magnitude of the variability itself), then SA will classify them as both highly sensitive to streamflow, with a sensitivity index close to one, regardless their robustness.

Consequently, the idea of conditioning robustness on SA results is unfortunately not applicable in the proposed framework. However, considering the value of the reviewer suggestion, we developed the 'limitation and future research direction' section accordingly as follows:

One of the main limitations of this study is the assumption of independence among the distributions used for the generation of the historical trajectories multipliers which constitute the foundation for developing the states of the world. Even though such assumption allows to explore the full range of variability of the exogenous drivers of the system, and has been successfully applied in recent hydrological applications (see for example Pianosi and Wagener 2016 and Amaranto et al. 2020), a study entangling the covariance among the uncertainty sources could provide further insights on the robustness of each operating policy, and can tailor the sensitivity analysis on a more reliable set of perturbations. It is therefore a recommendation for a future research exercise. In this study, the sources of vulnerability and the uncertain output realizations of the optimal water supply strategies are investigated only for robust policies. Therefore, RA and SA are sequential methodological steps rather than being interdependent. Future research could be developed towards implementing SA methods that establish feedback loops with RA, in order to enhance the robustness of operating policies against uncertain exogenous factors. Furthermore, robust policies are selected here according to the *minimax* robustness metric. *minimax*, by identifying the best alternative in the worst possible input realization, encompasses a risk-adverse behavior of

the decision maker. However, other criteria are available in the literature (see Giudici et al. 2020 for a review), each of them representing a different level of risk perception and its associated definition of robust operating policy. For a more comprehensive analysis, one could include additional robustness metrics in the experimental setup and evaluate how the choice of the metric shapes the selection of robust policy and the identification of the sources of vulnerability thereof. A possible architecture in this regard can be found in Herman et al., (2015), where four different robustness definition are considered, and robust planning decisions are developed accordingly

**Specific comments**

**RC:** *the model description may not be immediate to a reader not used to such models. It would be good to make clear what are the policies, the uncertainties etc. etc. - a good possible framework to follow is the XLRM (actually linked to RDM)*

**AR:** We thank the reviewer for the precious suggestion. To clarify, we have expanded section 3.3, and framed our approach within the XLRM.

Section 3.3 was modified as follows:

According to Hermann et al., 2015, a robustness analysis is usually carried out by performing the following sequential steps: (1) generation of alternative policies; (2) sampling of possible future scenarios and; (3) computation of robustness metric via system re-simulation. Table 3 provides a conceptualization of the three aforementioned steps tailored upon this study, following the well-known XLRM framework (Lempert, 2003). In the framework, X are the exogenous uncertainty sources; L (of lever) are the different alternative water management strategies (i.e., the policies identified via optimization) to be explored; M (of measure) refers to the performance metrics used to rank the desirability of the different policies (L) in the face of the exogenous uncertainties (X); and, finally, R refer to 'relationships in the system' (i.e., the model), which define how the exogenous uncertainties (X), policies (L) as well as outcomes (M) are tied together and relate to each other (Ciullo et al., 2019).

Table 2. Conceptualization of the robustness analysis implemented in this study

| Uncertain Factors [X] | Policies [L] |
|---|---|
| Climate | |
| Irrigation demand | Optimal operating policies found via optimization |
| Additional inflow from Corumana dam | |
| Urban Water demand | |
| **Relationships [R]** | **Performance Metrics [M]** |
| *Umbeluzi model* | *minimax* robustness metric |

**RC:** *better description of the GLUE method*

**AR:** We thank the reviewer for the comment. The description of the GLUE method required indeed some clarifications. The text has therefore been edited as follows (section 3.4.1):

We perform the quantification of the output variability in response to the four uncertainty sources considered in the paper by employing the GLUE method. In particular, GLUE

allows for determining which SOW lead the optimal robust policies to yield unacceptable results. The implementation of GLUE encompasses several steps, most of which are already included withineither the robustness or the sensitivity analysis. A brief summary is provided below:

1. Generation of the States of the World: as described above, SOWs are obtained in this study by near-random sampling of the perturbed time series of inflow, irrigation demand, infrastructure and population.
2. Specification of the  objective function: this function is defined in this study as each of the four objectives which constitutes the four-dimensional objective function in Equation 10b.
3. Definition of the threshold values for the behavioral system perturbation set: here, the threshold value is defined for each operating policy as the 5th percentile of all the objective function realizations, after simulating the system for Nu SOW. The behavioral perturbations are defined as the multiplier's values keeping the objective function below the threshold.

From a computational perspective the GLUE algorithm requires, for a certain robust operating policy, to re-simulate the system for each $\chi \in \Xi$ (i.e.: for each of the states of the world generated during robustness analysis). At every simulation, the value of the objective function is computed and stored. Upon completing this step, an empirical cumulative distribution function is fit to each of the four objective function's dimensions (i.e.: the operating objectives). Then, the algorithm checks whether a state of the world leads or not to an objective value below a certain threshold (which in this study is set as the 5th percentile of the objective CDF). In the former case, the SOW (and the system perturbation set the SOW is constituted by) is classified as behavioral, in the latter as non-behavioral. Considering the expected adverse effect which most of the uncertain future realization of the external system drivers might have on water availability and demands, the choice of the 5th percentile as threshold value is representative of the stakeholder desire to stay as close as possible to the historical performance.

**RC:** *when introducing RID and RUD (at about line 380) it isn't immediately clear that the considered least robust alternative is not the asbolute least robust, but rather the least robust among the Pareto set (i.e., performing best for at least one stakeholder).*

**AR:** The distinction between RID and RUD have been clarified as follows:

Figures 7a and 7b represent a scatterplot of the downstream irrigation objective in correspondence with its relative forcing perturbation for RID (most robust alternative for downstream irrigation) and RUD (least robust alternative among those performing best for at least one stakeholder), respectively.

**RC:** *Also, it would be ideal trying to condense the info from figures 7 to 9 into one figure - or at least plotting them together as they are "the same".*

**AR:** We thank the reviewer for the suggestion. It is indeed true that figures 7 and 9 refers both to the irrigation sector. However, downstream and upstream irrigation are treated as two separate stakeholders, since they are often in conflict with each other, and. As a consequence, each individual stakeholder necessitated a tailored upon description of their conflicts and their inherently different vulnerability to exogenous changes across policy, which inherently different. To facilitate such description, we have decided to keep the two figures separated.

**RC:** *better captions + some remaining typos/language issues*

**AR:** we improved the caption and checked spelling and grammar through the document.

**Reviewer 2**

*Review of "Disentangling Sources of Future Uncertainties for Water Management in Sub-Saharan River Basins", Amaranto et al. This paper presents a study that applies a decision framework for developing optimal operating policies for the Barragem de Pequeños Libombos (BPL) Reservoir in Mozambique. This reservoir has been constructed to serve the water supply to the greater Maputo area. It also serves irrigation up and downstream of the reservoir, and a hydropower station to generate energy. A multi-objective evolutionary optimisation approach is applied to find optimal operating policies. The robustness of these policies to changes in irrigation demand, urban demand, and climatic conditions is then explored. Finally, the sensitivity of both robust and non-robust policies to these changes is explored, concluding that robust policies are sensitive primarily to changes in climatic conditions and consequent changes in reservoir inflows, while non-robust policies are also sensitive to population growth and structural interventions. This last points is perhaps one of the more interesting aspect of the contribution to the readership of HESS. Besides that the paper seems to focus primarily on the application of the optimisation approach and exploration of robustness and sensitivity. In this sense the scope of the paper appear to hinge on two thoughts, one being the optimisation approach that is proposed and in particular the exploring the sensitivity of optimal polices, and the second the application to the case of the BPL reservoir. Though the application of the methodology is of interest, and to some extent of an interest to the readership of HESS, the manuscript does not reflect on the findings in the broader context than the particular case study in which the method is applied. As a result it is also not entirely clear to what extent the conclusions established are case study specific, or if these can be made more generically. This would require a much improved discussion interpretation of the results. This would also be more appealing to the scope of HESS if that discussion is presented within the hydro-sociological context and how the specific choices and assumptions that are made reflect on the generality of the conclusions. In the general comments below this is elaborated on. Overall the paper is well written and well structured. However, the authors have not really paid much attention to the correct use of grammar. While a possible copy-editing phase would easily solve several issues, a more careful preparation would have been expected. There are many sentences with multiple grammatical errors. Some examples are included in the detailed comments, though these are not considered to be exhaustive but rather as illustrative.*

**AR:** We thank the reviewer for this outstanding review. Her/his constructive comments helped us improve the manuscript and strengthen our analysis.

**General Comments**

**RC:** *It would be very helpful if more details are provided on the hydrological characteristics of water availability and demand, and also reflecting on the size of the reservoir in relation to that demand. First, it would be useful to be explicit as to the average availability of water. A rough estimate made from figure 2 would suggest that the mean annual inflow is about 7.5 m3 /s, with 10-90 percentile variability as indicated. This constitutes about 236 Mm3 per year. The volume of the reservoir is reported to be 382 Mm3 per year. This would then be 50% more than the average annual inflow, which means that for this reservoir the inter-annual variability is of importance. The paper does mention that a part of the storage is inactive, which also influences the sensitivity of the policies, but it is not clear how much this inactive storage is. Reviewing the numbers pertaining to demand, the irrigation demand (upstream and downstream) is mentioned to be some 33 Mm3 per year on average. The water supply demand is stated to be 80 Mm3 so it is clearly the dominant demand, which is in line with the purpose of the reservoir (it would be useful to understand the date for which this demand has been established). Demand to meet environmental needs is set at 15%, with a simple estimate being*

*about 35 Mm3 . In any case, these numbers suggest that total demand (irrigation + urban + environmental) used as a benchmark is about 62% of the average annual inflow. It is also some 38% of the storage of the reservoir. Though these percentages have been estimated somewhat roughly based on the data, they are useful to interpret the results, to understand the degree of water stress, and also how well the reservoir can buffer the variability of the inflow to meet demands. I believe the authors should develop a more extensive reflection on how the results they find depend on these ratios of availability and demand, as a more stressed situation where availability and demand could well yield quite different results. It would also be important to understand the inter and intra annual variability of the inflow, related to these ratios, and a more elaborate discussion of how these ratios influence the conclusions on robustness and sensitivity. What would happen if, for example the available storage of the reservoir was much smaller than the average annual inflow, or the average annual demand? The numbers presented as a baseline do not suggest a situation that is under extreme stress. Also, the changes to irrigation demand (25%) and Urban demand (2%) constitute in absolute values only some 8 Mm3 and 1.6 Mm3 annually. This appears to be somewhat modest in relation the average annual inflow and the size of the storage, which would suggest variability and uncertainty can be easily buffered. I would be very curious if the authors could reflect on how these values influence their conclusions on the importance of the variability of the hydrological inflows which suggest a reduction of between 5% and 40% of the inflow, constituting some 12 to 95 Mm3 , so already much more dominant from the outset. The manuscript would increase in scientific interest if a more elaborate discussion is provided that reflects on the ratios between the available water and the demands, as well as the relative magnitude of variability and changes.*

**AR:** We thank the reviewer for the very useful comment and the detailed explanation. Even though hydropower does not represent a consumptive water use, the turbine capacity exceeds downstream irrigation and urban demand combined. Once water is released from the reservoir and turbined to produce electricity, part is diverted to the irrigation district, part is used for urban supply, and the reminder flows in the estuary of the river, exiting therefore from the system under investigation. Consequently, the total water consumption might exceed the 62% of the average annual inflow. This, in turns, might cause water shortages during the winter season, when the inflow reaches its annual minima.

This is confirmed by the severe water scarcity issues affecting the Pequenos Libombos reservoir, which stood at about 18% capacity in January 2021, and 26% in January 2020 (similar issues were also recorded regularly, through the past decade). It is therefore reasonable to assume that the system under the baseline conditions is indeed under extreme stress.

However, we agree with the reviewer suggestion: it would be useful for the reader to provide more details on the hydrological characteristics of water availability and demand, contextualizing such trajectories on the size of the reservoir and the dynamics of storage though the year. Therefore, in the revised manuscript we will better clarify the water quantity involved and their relationship as follows (section 2.1):

> The Umbeluzi river flows across three countries (South Africa, Swaziland, and Mozambique), draining an area of about 5400 $km^2$ (Fig 1a) before discharging in the Indian Ocean through the Espirito Santo Estuary south of Maputo.
>
> The hydroclimatic regime is subtropical, with hot and wet summers from November to May, followed by dry, warm winters from June to October (Figures 1a and b). The average annual streamflow in the lower Umbeluzi (regulated through the Mnjoli Dam in Eswatini, in the upstream part of the river basin) is of about 220 $Mm^3$ (corresponding to 7 $m^3 s^{-1}$) and, following the hydroclimatic pattern, is unevenly partitioned through seasons, with 78% of

the total discharge occurring in summer (171 $Mm^3$, corresponding to 9.3 $m^3 s^{-1}$) and the reminder 22% in winter (49 $Mm^3$, 3.75 $m^3 s^{-1}$). As far as the inter-annual variability is concerned, the Umbeluzi river basin is characterized by frequent prolonged droughts, with the average inflow reduced to only 119 and 100 $Mm^3$ in 2007 and 2015, respectively. Excluding the year 2000, where a terrible flood hit Mozambique causing about 800 fatalities, the wettest year among those on record (2006-2016) was 2010, when the total discharge reached about 360 $Mm^3$.

About 40 km upstream the Estuary, the river flows into the Pequenos Libombos Dam (BPL). With The dam has a storage capacity (including 10.2 $Mm^3$ of inactive storage) of 382 $Mm^3$. BPL was constructed in 1987 with the goal supplying water to the metropolitan area of Greater Maputo (including the municipalities of Maputo, Matola and Boane), especially during winter, when the system is often exposed to droughts. Other operation targets include hydropower generation and irrigation supply, both upstream and downstream the reservoir. Upstream irrigation districts extend for about 2500 ha, and have an yearly water demand of 22.8 $Mm^3$ (Figure 1c). They abstract water directly from the reservoir to grow mainly tropical fruits (mango and bananas). Water is discharged from the dam into a power plant with a capacity of 1.8 MW. After flowing through the turbine, reservoir releases and then serve both urban and irrigation (bananas) demands, which are estimated around 80 and 11.5 $Mm^3 y$ (Figure 1c), respectively. To preserve ecosystem sustainability, a minimum flow constraint corresponding to 15% of the cyclostationary monthly inflow is imposed in the Estuary. A summary of the main hydroclimatic patterns across seasons, including aggregated values of water demands by sectors, is provided in Table 1.

Table 1: Summary of the main hydroclimatic variables and water demand sources aggregated by sector

|  | Total | Summer | Winter |
|---|---|---|---|
| Inflow | 220 | 171 | 49 |
| Upstream irrigation | 22.8 | 12.1 | 10.7 |
| Downstream irrigation | 11.5 | 6.1 | 5.4 |
| Environment | 33 | 26.65 | 7.35 |
| Urban | 80 | 46.6 | 33.4 |

Furthermore, additional information on the physical constraints shaping the interdependence between the release decision and the actual release have been provided in section 2.2 as follows:

The latter are in fact constrained by physical constraints (i.e., spillway activation and inactive storage threshold) within a discretionary operating space by the maximum and minimum feasible release function (see Soncini Sessa et al., 2007 for more details). In particular, the minimum release function constrains the release to zero in case the available volume in the reservoir $s_t$ equals the inactive storage (10.2 $Mm^3 s^{-1}$), while the maximum hydraulic outflow $r_{t+1}^{MAX}$ can be formulated as follows:

$$r_{t+1}^{MAX} = 4.6(h_t - 15.25)^{0.5}$$

In addition, to provide the reader with the proper set of information about how the sustainability of the different policies might be shaped by future patterns in water availability and demands, we have expanded section 3.3.1 with a table with the descriptive statistics of the variability range of the various uncertain exogenous driver. The section has been modified as follows:

As a result, K= 10000 deeply uncertain inflow, irrigation demand, additional inflow from Corumana and urban demand scenarios are extracted. The set Ξ including all the states of the world where the operating policies are evaluated is generated with a Latin Hypercube Sampling (LHS) across the four dimensions of the uncertainty sources, with a sample size Nu= 5000(cardinality of Ξ).

Table 2 provides a summary of the descriptive statistics of the SOWs aggregated by uncertainty source, including the driest and wettest year, as well as the maximum and minimum water demand, and the maximum and minimum yearly inflow from Corumana.

Table 2: statistical extremes of the SOWs, aggregated by uncertainty source

| Streamflow $Mm^3y^{-1}$ | | |
|---|---|---|
| Historical average | Driest year | Wettest year |
| 220 | 60 | 342 |
| Upstream irrigation $Mm^3y^{-1}$ | | |
| Historical average | Minimum demand | Maximum demand |
| 22.8 | 22.8 | 28.5 |
| Downstream irrigation $Mm^3y^{-1}$ | | |
| Historical average | Minimum demand | Maximum demand |
| 11.5 | 11.5 | 14.4 |
| Urban demand $Mm^3y^{-1}$ | | |
| Historical average | Minimum demand | Maximum demand |
| 80 | 80 | 98 |
| Inflow from Corumana $Mm^3y^{-1}$ | | |
| Historical average | Minimum yearly inflow | Maximum yearly inflow |
| 0 | 0 | 56 |

We believe that this will also provide insights for a better framing of the results regarding the sensitivity of the operating policies to uncertain exogenous factors.

**RC:** *I have some doubts on the formulation of the optimisation problem, and how representative this is for the case being studied. It appears that the water supply to Greater Maputo is considered a lower priority to meeting irrigation demand. There is a hedging rule applied in Eqn 3, but ultimately the water supply of Maputo is established as the remainder of the actual downstream release, after actual delivery to irrigation and meeting the environmental flow demand (eqn.5). In the introduction, however, it is noted that the reservoir is currently operated in a different way. The supply to Maputo gains priority over irrigation, with the latter being stopped if there is insufficient water. This suggests a fundamentally different policy is followed than that which is suggested here. This raises the question how realistic the proposed approach in modelling the operating policy is, and what the impact of this limitation is on the findings and conclusions.*

**AR:** The reviewer is correct. Meeting demand for the city of Maputo is pillar in the operation of the Pequenos Libombos reservoir. However, in the definition of the optimization problem, equation 3 does not prioritizes irrigation over urban supply. In fact, equation 3 defines the

fraction of releases to be diverted for irrigation purposes as inversely proportional to $\alpha$, and exponentially growing with respect to $\beta$.

The feasibility set of such parameters allows the evolutionary algorithm to also explore $\alpha$ and $\beta$ values (and combinations thereof) which places urban supply in foreground with respect to irrigation. It is (as an example) the case of $\alpha$ values much higher than the reservoir releases (the $\frac{r_{t+1}^d}{\alpha}$ term would assume near zero values) or the case of large $\beta$ values for $\frac{r_{t+1}^d}{\alpha} < 1$ (the amount of water diverted to irrigation drops to about 15% of the demand for $\frac{r_{t+1}^d}{\alpha} = 0.9$ and $\beta = 20$, and to 1% of the demand for $\beta = 45$). The opposite would be true for $\alpha$ values considerably smaller than the release decision.

In other words, the optimization problem formulation generates $\alpha$ and $\beta$ combinations which could allow the exploration of the whole irrigation-urban supply tradeoff. We are aware that, in the actual operation of the Pequenos Libombos reservoir, the supply to Maputo gains priority over irrigation, with the latter being stopped if there is insufficient water. However, to analyze any possible behavior of the regulator, we decided to include also operating policies which prioritize irrigation over urban demand.

Considering the importance of a clear understanding of the operating rules which characterize the diversion dam, we have provided a more detailed explanation of equation 3 in the revised manuscript (section 2.2) as follows:

> Equation 3 defines the fraction of releases to be diverted for irrigation purposes as inversely proportional to $\alpha$, and exponentially growing with respect to $\beta$. It follows that, according to the values assumed by such parameters, urban supply could (as it occurs, for example in case of $\alpha > r_{t+1}^d$ or for $\beta > 1$ when $\frac{r_{t+1}^d}{\alpha} < 1 < 1$) or could not ($\alpha < r_{t+1}^d$) be placed in the foreground with respect to irrigation. Even though this does not correspond to the actual operating rule of the diversion dam (which systematically prioritizes urban supply in case of water scarcity conditions), this study aims at exploring the whole irrigation-urban supply tradeoff. Therefore, equation 3 is set such that also irrigation-prone solutions could be discovered.

**RC:** *I would expect that a much higher sensitivity to population growth would then be found, and that the competition between urban supply and irrigation would be exacerbated, as in more extreme growth population growth scenarios this prioritisation would mean irrigations may become unfeasible. I would again think the manuscript would gain in interest if a discussion is provided as to the extent to which the formulation of the optimisation problem reflects the actual operation of the reservoir, as well as how that formulation influences the conclusions found.*

**AR:** We thank the reviewer for the precious comment. The sensitivity to population growth strictly depends on the operating policy upon which the sensitivity index is calculated. For example, Figure 8 shows a negligible impact of population growth on the urban deficit for RUD (blue dots and boxes), while the opposite is true for NR (red dots and boxes).

In addition, Figures 5 and 6 evidence how the competition between urban supply and irrigation is indeed exacerbated in the most extreme scenarios, with RID and RUD providing similar objective function values in the baseline but penalizing in turns one stakeholder over the other under deep uncertainty conditions (i.e.: in water scarcity conditions).

The low sensitivity values of irrigation supply with respect to population growth, (as well as the high sensitivity of irrigation to agricultural area expansion) are depending upon the formulation of the operating rules for the diversion dam expressed in equation 3 (i.e., release from the dam and agricultural demand). However, figure 7 clearly show that the hypothesis of 'feasible irrigation no matter the population growth' holds only for expansion in irrigated area up to about 2%, while for any further expansion the operating rules of the diversion dam prioritize urban supply causing a strong deficit increase. This is also confirmed by figure 8a, which shows how RUD, by diverting water to the city of Maputo, can fulfill urban demand no matter the population growth rate, penalizing irrigation as a consequence (if the streamflow depletion is lower than about 35% and the construction of the pipeline is completed soon enough).

Once again, the reviewer is correct in pointing out that the supply to Maput would always gain priority over irrigation in a real-world situation, especially in the most extreme conditions. For the sake of providing the full spectrum of tradeoff, we have included all the possible operating options in our analysis. To ensure that the reader has the complete set of information, we have clarified this aspect in the revised manuscript as follows (see also comment above):

> Equation 3 defines the fraction of releases to be diverted for irrigation purposes as inversely proportional to $\alpha$, and exponentially growing with respect to $\beta$. It follows that, according to the values assumed by such parameters, urban supply could (as it occurs, for example in case of $\alpha > r_{t+1}^d$ or for $\beta > 1$ when $\frac{r_{t+1}^d}{\alpha} < 1$ <1) or could not ($\alpha < r_{t+1}^d$) be placed in the foreground with respect to irrigation. Even though this does not correspond to the actual operating rule of the diversion dam (which systematically prioritizes urban supply in case of water scarcity conditions), this study aims at exploring the whole irrigation-urban supply tradeoff. Therefore, equation 3 is set such that also irrigation-prone solutions could be discovered.

**RC:** *Related to this comment is also the question on how realistic the scenarios chosen are. The growth rate of Maputo has been set at some 2%, which seems modest. Those familiar with the area would know that Matola is growing much faster than Maputo, at some 4% per year, and is already larger than Maputo itself.*

**AR:** The reviewer is correct: Matola is growing much faster than Maputo. The growth rate of 2% per year which is used in this study is representative of an average over the whole area: the metropolitan area of Greater Maputo which includes the Municipalities of Maputo, Matola, and Boane, and it is provided in the literature by:

*"Droogers, P., de Boer, F., and Terink, W.: Water Allocation Models for the Umbeluzi River Basin, Mozambique, 2014, Report Future Water 132, Fryslân".*

**RC:** *Also, can the growth in demand be considered as linear with population? Several studies show that increasing development also result in an increase in water demand disproportional to population growth. Given the dominance of the urban demand, and the very modest increase such factors could well be discussed. It may be that the conclusions found on the sensitivity of robust strategies are indeed valid within the very modest growth scenarios explored. There may well be a limit to the change in which the conclusion on sensitivity holds, after which this sensitivity may increases dramatically. This would help generalise the findings of the paper.*

**AR:** we thank the reviewer for the suggestion. It is true, we assumed an irrigation demand which grows linearly with population. However as in *Droogers* et al., (2014), the demand grows overall exponentially, considering an exponential population growth of 2% every year.

The reviewer is right: the sensitivity of robust strategies holds within the perturbation set we assumed for this study. This is true for population, as well as for irrigation expansion and streamflow depletion. We have clarified this aspect in the conclusion as follows:

> The aforementioned results in terms of UA and SA are specifically tailored upon the parametric input perturbation set employed to generate the SOWs. Even though each perturbation set can be well documented from the literature, unexpected changes in one of the exogenous factor (i.e. higher population growth or lower streamflow availability) could shape the behavior of the system, altering therefore the extent to which a certain policy is vulnerable to each uncertainty source. One of the main limitations of this study is the assumption of independence among the distributions used for the generation of the historical trajectories multipliers which constitute the foundation for developing the states of the world. Even though such assumption allows to explore the full range of variability of the exogenous drivers of the system, and has been successfully applied in recent hydrological applications (see for example Pianosi et al., 2016 and Amaranto et al., 2020), a study entangling the covariance among the uncertainty sources could provide further insights on the robustness of each operating policy, and can tailor the sensitivity analysis on a more reliable set of perturbations.

**RC:** *Continuing on from this comment, it would also be useful to understand the rationale of the objective functions themselves. Hydropower is clearly a by-product in the operation of the dam as there is no demand to meet (or a penalty if such demand is not met) and simply maximises the profit from generation. Given the primary purpose of the dam that would appear reasonable. The other objective functions, however, apply a squared residual to irrigation deficits and a linear residual to urban water supply deficit. This seems somewhat incongruous to already noted priority of urban water supply to irrigation, as it would tend to minimise irrigation deficits with respect to deficits in urban supply. I agree that the factors in eqn.3 influence this preference, but it would be good to understand the rationale behind this choice. A more detailed description of the water allocation policy to irrigation would be useful. If there is insufficient water, and given that the crops are predominantly cash-crops, it would make sense that farmers simply scale the cropped area - which would suggest a linear loss if demands are not met.*

**AR:** The reviewer provides an interesting insight: it is indeed true that it would make sense that farmers simply scale the cropped area - which would suggest a linear loss if demands were not met. On the other hand, Hashimoto et al. (1982), suggests the use of a quadratic deficit to account for higher crop vulnerability to higher water shortages. We have clarified this aspect in the revised manuscript as follows (section 3.2.1):

> Where N (days) is the simulation horizon, $N_y$ are the number of years in the simulation horizon, $\beta$ a weight representing higher losses when the deficit occurs during the growing season. $W_t^{UP}$ and $r_{t+1}^{UP}$ are the irrigation demand and the amount of water pumped from the reservoir to upstream irrigation, respectively.
>
> The quadratic water supply deficit is a traditional formulation in reservoir operations since the work by Hashimoto et al. (1982). The square of the irrigation deficit accounts in fact for crop vulnerability by penalizing higher shortages, which are more likely to compromise the crop growth, with respect to more frequent but smaller shortages, which are less dangerous to the crops.

Further references on the use of the squared deficit as an irrigation step-cost can be found at:

Denaro, S., Castelletti, A., Giuliani, M., & Characklis, G. W. (2018). Fostering cooperation in power asymmetrical water systems by the use of direct release rules and index-based insurance schemes. Advances in Water Resources, 115, 301-314.

Giuliani, M., Li, Y., Castelletti, A., & Gandolfi, C. (2016). A coupled human-natural systems analysis of irrigated agriculture under changing climate. Water Resources Research, 52(9), 6928-6947

Hashimoto, T., Stedinger, J., Loucks, D. (1982). Reliability, resilience, and vulnerability criteria for water resource system performance evaluation. Water Resources Research, 18(1), 14–20.

**RC:** *Further discussion should also be provided on the scenarios developed to represent climate change. The method chosen is simple, which I agree is appropriate within the context of this study. However, if I understand correctly the change is a linear factor applied to the whole time series. This does not resolve any changes to the inter-annual variability of inflows. I am not fully familiar with climate outlooks for this particular basin, but presume the influence of climate change may be more multi-dimensional than a simple proportional reduction across the year. There could also be changes in the distribution of flow across the wet and dry season. Such shifts are of important to reservoir operation policies. This may be relevant to the ability of the reservoir to the meeting of irrigation demand, which is (logically) primarily in the mid and late dry season.*

**AR:** in our study, we adopted an approach resembling the delta change for the states of the world generation (Brown et al., 2012). The sampling strategy decreases the historical streamflow (which includes both dry and wed hydrological conditions) by as much as 40%, following the results obtained by a conceptual hydrological model fed by nine downscaled general circulation models' results. The 5000 samples are drawn from a uniform distribution over the multiplier interval (0-0.6).

However, we agree with the reviewer. A change in the hydroclimatic patterns would most likely imply a shift in the hydrological regime of the river basin, with implications on the distribution of streamflow across seasons. Such variations in the streamflow distribution are not embedded in the so-generated states of the worlds.

On the other hand, the 110000 (10000 multiplier perturbations of the 11 years of historical data) hydrological years upon which the system is simulated provide, together with the samples from the other uncertainty source, a state of the world discretization grid which is dense enough to consider both the extremes and the intermediate scenarios over which the robustness of the various operating policies is computed, together with their sensitivity across uncertainty sources.

Following this line of thought, we modified the manuscript to provide the reader an understanding of the limitations of our approach in characterizing streamflow distribution through the year, and we enhanced the description on the scenarios developed to represent climate change as follows (section 3.3.1).

1. Climatic uncertainty: We generate high resolution scenarios of rainfall and temperature for the Umbeluzi river basin to the year 2100 by the quantile-quantile mapping (QQ-Mapping) downscaling procedure. We apply QQ-Mapping to coarse resolution data from three different Regional Circulation Models (ICHEC RCA4, ICHEC RACMO and ICHEC HIRHAM5, developed by the Swedish Meteorological and Hydrological Institute, the Royal Netherlands Meteorological Institute, and the

Danish Meteorological institute), simulated over three distinct representative concentration pathways: the RCP 2.6, RCP 4.5 and RCP 8.5 (Field, 2014). We use the resulting nine precipitation and temperature trajectories to force an HBV model (Lindström et al., 1997) validated over the control period, generating nine inflow trajectories. We then use the minimum (0.05) and the maximum (0.4) of the projected percentage inflow decrease as the feasibility set boundary of a uniform distribution, from which we extracted K= 10000 inflow multipliers in the interval [0.6,0.95] using simple random sampling. As a result, the historical inflow (which includes both dry and wet hydrological conditions) is decreased in the different scenarios by as much as 40% (as the product between the historical trajectories and the multipliers). Such methodological procedure for the generation of the states of the world is often referred to as the delta method (Brown et al., 2012), and has been widely used in the literature (see for example Bertoni et al. 2019). One of its main drawbacks is not being able to account for a seasonal shift in the hydrological regime which would naturally follow a change in the hydroclimatic patterns. However, the 110000 (10000 multiplier perturbations of the 11 years of historical data) hydrological years upon which the system is simulated provide a states of the world discretization grid which is dense enough to consider both the extremes and the intermediate scenarios over which the robustness of the various operating policies is computed.

"Brown, C., Ghile, Y., Laverty, M., & Li, K. (2012). Decision scaling: Linking bottom-up vulnerability analysis with climate projections in the water sector. Water Resources Research, 48, W09537. https://doi.org/10.1029/2011WR011212"

**Detailed Comments:**

**RC:** *Please revise the grammar in the whole manuscript. Here is an example of one or two sentences with multiple grammatical errors. All minor and easily resolved, but a careful grammatical review could improve the readability (green indicates and addition, red-strikethrough a deletion, while text in blue indicates where the text does not make sense, grammatically).*

*Lines 33-38 An archetypal example of a highly regulated, fast evolving South-Saharan hydrosystem is the Lower Umbeluzi river basin, Mozambique. About 45 km upstream of its delta in the Maputo bay, the river flows in the Barragem de Pequenos Libompos reservoir, which is operated to balance hydropower production, urban supply to the two million inhabitants of the Maputo province, and irrigation supply for to the 3600 ha of agricultural districts, mostly growing tropical fruits and sugarcane. A currently undergoing five year long drought have has boosted crop prices by about 50%, hindering food access to a population currently growing at rate of 0.6% per year and exacerbating conflicts among the urban, agricultural and energy sectors. currently undergoing: this is not grammatically correct as it is not clear what is currently undergoing a five year long drought.*

**AR:** We thank the reviewer for the suggestion. In addition to addressing the issues between lines 33-38, we have also revised the grammar through the whole document.

**RC:** *Line 38: Fryslân et al. 2014 is a very peculiar reference. In fact Fryslan is the organisation that published the report and is not a person. Either the organisation should be used, or the actual authors – but not a mix. The details of the reference in the reference list are insufficient and need to be completed. + Line 42: The world bank is funding*

**AR:** We have modified the text accordingly

**RC:** *Lines 44-47: This section describes that the policy makes have certain needs. How were these needs ascertained? Was this a discussion held with these stakeholders, or are the fact that these are needs based on an assumption made by the authors?*

**AR:** We thank the reviewer for the comment. However, lines 44-47 does not imply certain needs by decision makers. Instead, it is highlighted how, considering the uncertain hydroclimatic and social evolution of the system, it is reasonable to assume that the design of optimal operating policies which are also robust against such uncertain drivers might assist policy makers towards making informed decisions.

**RC:** *Line 51: I am not sure this paper can be said to be tackling the evolution of the system drivers. Perhaps identifying would be a better word.*

**AR:** The manuscript was modified according to the reviewer comment.

**RC:** *Line 71: what does not seem the correct word – please rephrase.*

**AR:** The sentence was rephrased as:

> Our framework complements  the findings by Quinn et al. (2019) by investigating the role of uncertain exogenous drivers in shaping the effectiveness of optimal operating policies to sustain the agricultural, urban and energy sectors.

**RC:** *Lines 80-88. Please provide more detail on the dam and its outlets. In particular the size of the inactive zone (see general comments) and the maximum capacity of the penstock, and if the penstock is the only controlled outlet structure. This is suggested to have a capacity of 2x the combined downstream irrigation and urban demand, which would be 2.9 m3/s. Back-calculating with an installed capacity of 1.8 MW this would suggest the turbine efficiency is 73%, which is quite low. Please provide the correct numbers to help interpret the various results. In several places in the text it is suggested these are known, but they do not appear to be explicitly mentioned. Additional discussion should also be included on the presence of upstream reservoirs in the basin in Eswatini. These do exist.*

**AR:** We thank the reviewer for the precious comment. In line to what is stated in the general comments, we have expanded the study area description (see author's response to general comment 1), providing all the available and sharable information on the dam.

**RC:** L *80: upstream of the estuary + Line 81: goal of supplying + Line 82: in Line 36 it is mentioned that irrigation is used for tropical fruits and sugar-cane. Here bananas as mentioned (which are tropical fruits). What happened to sugar cane?*

**AR:** The manuscript was modified according to the reviewer comment.

**RC:** *Figure 2a. I do not fully understand the red line in this figure, but assume this is the median daily inflow. However, there are several peaks where the red line is at the limits of what the caption suggests is the 10-90 percentile range. Also, during the dry season, there seems to be an almost constant maximum to the 10-90 percentile range. Please clarify these data as these are if I understand correctly simulated data from the HBV model. Figure 1a. The upstream irrigation areas in the map are incorrectly coloured. Also, the figures show there are unregulated tributaries. To what extent do these have a role in meeting demand?*

**AR:** We agree with the reviewer, the interpretability of Figure 2 regarding inflow can be enhanced. Observed inflow data were provided by by The Administracão Regional de Ãgua (ARA) Sul with a daily time step for the time-period 2006-2016. In order to facilitate the understanding of the Umbeluzi hydrological regime, we have modified the figure by introducing the cyclostationary average with 10 days moving window (blue line), combined

with the 10th-90th interquartile, computed within the same moving window. The updated version of Figure 2 is reported below:

[Figure]

**Figure 2.** cyclostationary average of the main hydroclimatic variables computed over a 10-days moving window. (a) Precipitation and streamflow; (b) temperature; (c) downstream irrigation demand; and (d) upstream irrigation demand. The shaded areas represent the 10-90th interquartile range for each variable.

As far as Figure 1a is concerned, the color of the upstream irrigation districts was modified to match Figure 1b, and now it appears as follows:

[Figure]

**Figure 1.** (a) Study area. (b) Topological map of the system

**RC:** *Line 106: I assume the water is pumped to the irrigation districts and not from. + Line 129: Check spelling*

**AR:** The manuscript was modified according to the reviewer comment.

**RC:** *Line 177/179: Please clarify the units for hydropower. Also, please provide the value of the efficiency of HP production applied.*

**AR:** The units for hydropower [MWh] and the turbine efficiency (70%) have been added to the manuscript

**RC:** *Line 199: the sentence "which are known for well-validating out of sample data" needs to be rephrased as it is not clear*

**AR:** The sentence has been rephrased as:

> which are known for their generalization ability and robust performances in validation

**RC:** Line 234-260: The description of the samples is not clear. It would be clearer to describe each set of samples as a vector that is made up of the four parameters that sampled from their respective (uniform) distributions. Then K-vectors are sampled.

**AR:** In order to clarify the description of the samples, the text was modified as follows:

> As a result, K = 10000 deeply uncertain SOWs embedding inflow, irrigation demand, additional inflow from Corumana and urban demand scenarios are  generated. Each of the K set could be seen as a vector composed by four parameters (i.e., one multiplier for each uncertainty source), sampled from their uniform distribution. The set $\Xi$ including all the states of the world where the operating policies are evaluated is generated with a Latin Hypercube Sampling (LHS) across the four dimension of the uncertainty sources, with a sample size Nu= 5000 (cardinality of $\Xi$).

**RC:** *Line 254: What is value of the project horizon H?*

**AR:** The sentence has been rephrased as:

> The completion day cd of the project is treated as a stochastic variable, for which K samples are drawn from a uniform discrete distribution in the interval [0, H], where H  = 11 years (i.e., the simulation horizon for an individual state of the world)

**RC:** *Line 286: Please be specific on the likelihood function used. Is this constructed based on the simple summation of the four constituent objective functions? Please clarify how these are weighted. + Line 289: What is the rationale of selecting the 5%? Is there any relationship to impacts? Also it is not completely clear if this is the 5th exceedance of 5th non-exceedance percentile. Please clarify.*

**AR:** We thank the reviewer for the precious comment. In order to tailor on each stakeholder the likelihood of a certain policy to yield acceptable performance, the four objectives constituting the objective function have not been aggregated. To clarify, this aspect, and the rationale behind selecting the 5[th] percentile threshold, we have expanded the description as follows:

We perform the quantification of the output variability in response to the four uncertainty sources considered in the paper by employing the GLUE method. In particular, GLUE allows for determining which SOW lead the optimal robust  policies to yield unacceptable results. The implementation of GLUE encompasses several steps, most of which are already included within either the robustness or the sensitivity analysis. A brief summary is provided below:

1. Generation of the States of the World: as described above, SOWs are obtained in this study by near-random sampling of the perturbed time series of inflow, irrigation demand, infrastructure and population.
2. Specification of the  objective function: this function is defined in this study as each of the four objectives which constitutes the four-dimensional objective function in Equation 10b.
3. Definition of the threshold values for the behavioral system perturbation set: here, the threshold value is defined for each operating policy as the 5th percentile of all the objective function realizations, after simulating the system for Nu SOW. The behavioral perturbations are defined as the multiplier's values keeping the objective function below the threshold.

From a computational perspective the GLUE algorithm requires, for a certain robust operating policy, to re-simulate the system for each $\chi \in \Xi$ (i.e.: for each of the states of the world generated during robustness analysis). At every simulation, the value of the objective function is computed and stored. Upon completing this step, an empirical cumulative distribution function is fit to each of the four objective function's dimensions (i.e.: the operating objectives). Then, the algorithm checks whether a state of the world leads or not to an objective value below a certain threshold (which in this study is set as the 5th percentile of the objective CDF). In the former case, the SOW (and the system perturbation set the SOW is constituted by) is classified as behavioral, in the latter  as  non-behavioral. Considering the expected adverse effect which most of the uncertain future realization of the external system drivers might have on water availability and demands, the choice of the 5th percentile as threshold value is representative of the stakeholder desire to stay as close as possible to the historical performance.

**RC:** *Line 371-374: It is somewhat confusing to refer to hydropower having a demand in the context here. Hydropower is generated as a by-product of the release to downstream irrigation and to meet urban demand. I would rather rephrase it as hydropower production.  + Line 378-379: The grammar of this sentence is very poor. Please improve.*

**AR:** We thank the reviewer for this comment. The grammar of the sentence has been adjusted, and the text in lines 371-374 has been modified according to the reviewer's comment.

**RC:** *Line 387-388: Could the ability to meet the RID to endure water supply no matter the expansion of the irrigation area be linked to the size of the area. The maximum change of the d/s irrigation is on the order of 2.7 Mm3 which is quite a limited amount (~3.4%) compared to the benchmark public water supply (see also general comments)*

**AR:** The reviewer is correct. All the results presented in this study refer to the parametric input perturbation set employed to generate the SOWs. This aspect has been clarified in the conclusion of the manuscript.  Furthermore, we have modified the sentence in lines 387-388 as follows:

In particular, irrigation deficit is produced only when the streamflow multiplier falls below 0.65. This practically means that, for any future climatic conditions generating streamflow reduction below to 35%, RID is able to ensure water supply no matter the expansion (among those embedded in the SOWs) in irrigated area

**RC:** *Line 394: Please check the units (also in other lines). What does 140 m3 /s/Year mean? This would suggest this is the mean annual flow? That does not appear to be correct. Please check this and other similar units.*

**AR:** We thank the reviewer for this comment. Indeed, the squared component of the irrigation operating objective is missing. The correct value would be *140* $[(m^3s^{-1})^2Y^{-1}]$, and the manuscript has been modified accordingly.

The irrigation operating objective is obtained by applying Eq. 7-8, which computes the weighted squared deficit $[m^3s^{-1}]$ for each day of the simulation period. Such weighted squared deficits are then summed, and the resulting value is divided by the number of years, hence $[(m^3s^{-1})^2Y^{-1}]$. Similar applications can be found, among others, in Giuliani et al., (2016) and Denaro et al., (2018).

*Denaro, S., Castelletti, A., Giuliani, M., & Characklis, G. W. (2018). Fostering cooperation in power asymmetrical water systems by the use of direct release rules and index-based insurance schemes. Advances in Water Resources, 115, 301-314.*

*Giuliani, M., Li, Y., Castelletti, A., & Gandolfi, C. (2016). A coupled human-natural systems analysis of irrigated agriculture under changing climate. Water Resources Research, 52(9), 6928-6947*

---

## Author Response (AR2)

**General comment**

*I have reviewed the revised manuscript and would like to thank the author for their careful consideration of the comments by myself and the other reviewer. The manuscript has most certainly improved, and additional details provided contribute to the understanding. Despite the improvements the author have made, I am still struggling somewhat with the scope of the manuscript, given that it still hinges somewhat on two thoughts; the first being the optimisation of operating policies, robustness assessment and subsequent uncertainty analysis and sensitivity analysis of robust and non-robust policies; while the second is application to the Barragem Pequeños Limbobos (BPL). I think the conclusion section exemplifies this confusion, and that then is my main general comment on this revision. The bulleted conclusions appear to be primarily related to the findings of the application of the method proposed to BPL, and are therefore contingent on the particular structure of the studied water system, as well as already noted on the relative size of demands (e.g. the size of the demand from d/s irrigation when compared to the urban demand). This includes conclusions on the trade- offs between identified demands. The second part of the conclusion, which is in fact more of a discussion despite the name of the section, focuses more on the methods applied, but conclusions on the applicability or the generality of the method itself are lacking. This still brings to question how general the conclusions that are made are. In my opinion, perhaps one of the more interesting conclusions is that where the authors conclude that robust policies are vulnerable only to hydrological perturbations, whilst non-robust policies are also vulnerable to changes to urban and agricultural demand. That is an interesting conclusion if it is indeed generic and therefore a clear contribution to the establishing of reservoir operating polices through the methods proposed. I would think that some discussion on how generic that conclusion is, or if it is indeed dependant on the configuration of the BPL water system. I would still call for the authors to add a short reflection on this. It may also be appropriate to label the section as conclusion and discussion. A separate discussion section prior to the conclusions would make the structure clearer, but I will leave this for the editor to decide on.*

**AR:** We thank the reviewer for this outstanding review. Her/his constructive comments helped us improve the manuscript and strengthen our analysis.

The scope of the manuscript is to propose a decision analytic framework for identifying the main sources of vulnerability to optimal-robust reservoir operating policies in multi-objective water management problems. Even though from a theoretical perspective such framework could be applied to any type of hydroclimatic regime, our study is tailored upon a specifically sub-Saharan river basin: the Barragem Pequeños Limbobos. Therefore, the conclusions thereof (i.e. robust policies are vulnerable only to hydrological perturbations, whilst non-robust policies are also vulnerable to changes to urban and agricultural demand) are to be considered contingent on the structure of the studied water system.

Even though it would be reasonable to assume that robust policies are intrinsically less affected by perturbation in uncertainty sources, the generality of such conclusion can be assessed only by testing test the proposed framework on another case study, with inherently different tradeoffs and water demand and availability trajectories. Notwithstanding the flexibility of the proposed framework towards the applicability on other river basin, such analysis would be beyond the scope of the manuscript.

However, considering the importance of clarifying upon the generality of the conclusions we have modified the final section of the manuscript as follows:

- Overall, it is possible to conclude that robust policies are usually vulnerable only to hydrological perturbations and are able to sustain the majority of population growth and agricultural expansion scenarios. Moreover, infrastructural interventions become crucial only in extreme drought conditions. On the contrary, non-robust policies are sensitive also to social and agricultural changes, and require structural interventions to ensure stable supply.

From a methodological and computational perspective, the proposed decision analytic framework could be easily applied to any type of water system. However, in this study it is customized for the BPL river basin, and so are the conclusions and discussions thereof. Therefore, a possible future research path could be directed towards testing the proposed methodology across a range of hydro

systems, in order to assess how stakeholder's tradeoffs, policy robustness and vulnerabilities are shaped by inherently different water availability and demand trajectories.

The aforementioned results in terms of UA and SA are specifically tailored upon the parametric input perturbation set employed to generate the SOWs. Even though each perturbation set can be well documented from the literature, unexpected changes in one of the exogenous factor (i.e., higher population growth or lower streamflow availability) could shape the behavior of the system, altering therefore the extent to which a certain policy is vulnerable to each uncertainty source.

**Detailed comments**

*Some detailed comments are provided below. The grammar of the revised version has indeed improved, though there are still some improvements that will need to be done. A good revision prior to submission of the revised manuscript would be recommended.*

*Line 82: Change reminder to remaining.*

*Line 86: I would change this to total discharge volume as that would be commensurate to the unit, which is a unit of volume.*

*Line 113: the closing bracket of the interval should be a square bracket.*

*Line 123: This constraint would appear (given its formulation) to be related to the capacity of the outlet through which releases are made. However, what confuses me is that the upstream release is direct from the dam itself (elsewhere it is suggested that it is pumped). So, does the constraint apply to the release to the upstream irrigation also? I am fine with it being simplified to consider releases as if these are effectuated through the same outlet, but at least make a comment to this as the optimisation space may be quite different if taking e.g. the max pump capacity upstream as the constraint.*

*Line 124: It is in turn and not in turns. This should also be corrected in one or two other places in the manuscript.*

*Line 141: I am not sure irrigation prone solution is a correct formulation. Perhaps this could be formulated as: 'solutions that favour irrigation can be discovered'*

*Line 174: yielding and not yielding to.*

*Line 304: The historical averages in the table for irrigation and urban demand are not, I presume, averages as determined over a time series (as was done I presume for the inflows). So, in a sense these are not averages as determined over a time series of demands I presume. Perhaps current or initial values would be a more appropriate label. Figure 2 suggest the demands were calculated based on extent the different crops and their evaporative demand as a function of climate data.*

*Line 323: This sentence does not make sense to me. Perhaps reformulate as below but do check the context: 'In particular, GLUE allows for determining which SOW result in optimal robust policies that yield unacceptable results.*

**AR:** we would like to thank the reviewer for the detailed review of the manuscript, we appreciate how addressing all the specific comments provided above will undoubtfully improve the overall quality and readability of the paper. Therefore, we implemented all the above suggestions while revising the document.